# Relaxin gene delivery mitigates liver metastasis and synergizes with check point therapy

Mengying Hu [1], Ying Wang[1,2], Ligeng Xu[1,3], Sai An[1], Yu Tang[4], Xuefei Zhou[1], Jingjing Li[2], Rihe Liu[2] & Leaf Huang[1]

Activated hepatic stellate cell (aHSC)-mediated liver fibrosis is essential to the development of liver metastasis. Here, we discover intra-hepatic scale-up of relaxin (RLN, an anti-fibrotic peptide) in response to fibrosis along with the upregulation of its primary receptor (RXFP1) on aHSCs. The elevated expression of RLN serves as a natural regulator to deactivate aHSCs and resolve liver fibrosis. Therefore, we hypothesize this endogenous liver fibrosis repair mechanism can be leveraged for liver metastasis treatment via enforced RLN expression. To validate the therapeutic potential, we utilize aminoethyl anisamide-conjugated lipid-calcium-phosphate nanoparticles to deliver plasmid DNA encoding RLN. The nanoparticles preferentially target metastatic tumor cells and aHSCs within the metastatic lesion and convert them as an in situ RLN depot. Expressed RLN reverses the stromal microenvironment, which makes it unfavorable for established liver metastasis to grow. In colorectal, pancreatic, and breast cancer liver metastasis models, we confirm the RLN gene therapy results in significant inhibition of metastatic progression and prolongs survival. In addition, enforced RLN expression reactivates intra-metastasis immune milieu. The combination of the RLN gene therapy with PD-L1 blockade immunotherapy further produces a synergistic anti-metastatic efficacy. Collectively, the targeted RLN gene therapy represents a highly efficient, safe, and versatile anti-metastatic modality, and is promising for clinical translation.

---

[1] Division of Pharmacoengineering and Molecular Pharmaceutics, Eshelman School of Pharmacy, University of North Carolina, Chapel Hill, NC 27599, USA. [2] Division of Chemical Biology and Medicinal Chemistry, Eshelman School of Pharmacy, University of North Carolina, Chapel Hill, NC 27599, USA. [3] Jiangsu Key Laboratory for Carbon-Based Functional Materials & Devices, Institute of Functional Nano & Soft Materials (FUNSOM), Soochow University, 215123 Suzhou, China. [4] Division of Pharmacotherapy and Experimental Therapeutics, Eshelman School of Pharmacy, University of North Carolina, Chapel Hill, NC 27599, USA. Correspondence and requests for materials should be addressed to L.H. (email: leafh@email.unc.edu)

Liver metastasis is the leading cause of death for patients with gastrointestinal (GI) cancers (e.g., colorectal cancer (CRC) pancreatic cancer) and some extra-GI cancers (e.g., breast cancer, lung cancer, and melanoma)[1]. While radical surgery is considered the only curative treatment currently, it is feasible for only 10–20% of patients with high recurrence rate due to the difficulty to eliminate residual metastases[2]. The chemotherapy regimen varies with respect to the primary tumor types[3,4] and provides limited survival benefit for liver metastasis patients due to the chemoresistance and off-target toxicity.

The metastatic colonization triggers liver fibrosis via the induction of profibrogenic factors. In turn, the fibrotic liver forms a favorable microenvironment for the engraftment and proliferation of metastatic tumor cells. Particularly, activated hepatic stellate cells (aHSCs), which represent the predominant mediator of liver fibrosis[5], promote chemotaxis and survival of tumor cells via CXCL12/CXCR4 axis[6] and various growth factors such as transforming growth factor-β (TGF-β) and hepatocyte growth factor[7]. Moreover, aHSCs shape an immunosuppressive milieu to protect metastatic tumors against immune surveillance by the production of cytokines such as IL-4 and IL-6[8,9], which promote the activation of M2 macrophages and myeloid-derived suppressor cells (MDSCs)[10] and induce T-cell anergy[11]. Excessively secreted extracellular matrix (ECM) by aHSCs also forms a physical barrier limiting T-cell mobility and penetration into the tumor area[12]. Indeed, despite success of immunotherapy in multiple cancer types, satisfactory therapeutic efficacy is rarely reported in liver metastasis. For instance, clinical studies demonstrated the association of liver metastasis with reduced response and shortened progression-free survival of PD-L1 checkpoint blockade therapy even for those originally well responding tumor types such as melanoma and non-small-cell lung cancer[13,14]. Therefore, targeting aHSCs not only represents a versatile strategy getting to the root of the problem for various liver metastases, but is also immunologically meaningful due to the potential to convert the cold tumor microenvironment to an immune sensitive milieu and hence synergize current immunotherapies.

Spontaneous regression of liver fibrosis after removing the etiology has been documented in both animal models and clinical situations[15]. However, the underlying physiological mechanism was rarely studied. Previous studies demonstrated a fibrosis-triggered expression of relaxin receptor family peptide-1 (RXFP1) in the liver and HSCs[16]. The binding of RXFP1 and its cognate ligand, relaxin (RLN), initiates the nitric oxide (NO) signaling against profibrogenic pathways[16–18]. Here, we further discovered hepatic RXFP1 expression undergoes dynamic changes during the disease development and regression, which peaks at the liver injury and gradually decreases at the resolution stage. Moreover, a baseline expression of RLN in the healthy liver is characterized, which increases rapidly during the fibrosis resolution. These phenomena suggest that RLN serves as an endogenous repair mediator reversing liver fibrogenesis. Moreover, hepatic RLN expression in the metastatic liver presents a gender-disparity pattern, with a higher level in females than in males. Concurrently, females display a significant slower progression of liver metastases in both rodent model and clinical situations compared with males, suggesting a possible role of RLN to confer females the antimetastatic privilege. Therefore, the regulatory mechanism of RLN to calm down aHSCs can be leveraged for antimetastasis treatment. Indeed, a very recent study has proved antitumor effect of RLN peptide in the pancreatic cancer model attributed to its stromal modulation function[19]. Despite the potential therapeutic efficiency, systemic administrated RLN usually displays compromised efficacy due to the short half-life (~10 min). The widely distributed RXFP1, particularly in the reproductive system[20], also reduces amount of the injected RLN arriving to the targeted site and induces possible side-effects.

To solve the problem, we sought to engineer an in situ depot capable of locally secreting RLN within liver metastatic lesions via targeted delivery of RLN plasmid (pRLN) by lipid-calcium-phosphate nanoparticles (LCPs). The nanoparticles are surface modified with aminoethyl anisamide (AEAA), a potent ligand for sigma-1 receptor (Sig-1R)[21], which is overexpressed by both aHSCs and metastatic tumors. We evaluated the RLN gene therapy on experimental murine CRC, pancreatic cancer, and breast cancer liver metastasis models via hemi-splenic inoculation of CT26-FL3, KPC, and 4T1 cells, respectively. The in situ enforced expression of RLN by transfected aHSCs and tumor cells potently inhibits metastatic tumor growth. The therapeutic effect is attributed to the reversion of aHSCs back to the quiescent phenotype and degradation of accumulated ECM. More importantly, the immunosuppressive environment within metastatic lesion is shifted to an immunostimulatory state with better cytotoxic T-cell infiltration after the treatment. The further combination of the RLN gene therapy with locally expressed PD-L1 trap fusion protein produced a synergistic antimetastatic efficacy in the CRC and pancreatic cancer liver metastasis models.

## Results

**An endogenous resolution mechanism for liver fibrosis.** Fibroblasts are recently recognized to be indispensable for tumor progression[11]. HSCs have been reported as the predominant source of intra-hepatic fibroblasts[5]. We first compared HSC-associated gene expression profiles (data extracted from the Gene Expression Omnibus (GEO; http://www.ncbi.nlm.nih.gov/geo) database under the accession number GSE68468) between liver metastasis lesions and matched normal liver samples from 40 CRC liver metastasis patients (Fig. 1a). Mesenchymal hallmarks mainly including α-smooth muscle actin (α-SMA; encoded by ACTA2 gene), fibroblast activation protein (FAP; encoded by FAP gene) and type I collagen (encoded by COL1A1 and COL1A2 genes), which are exclusively highly expressed in aHSCs[22,23], were upregulated by 4−16-fold in the liver with metastasis. Other ECM molecules such as periostin (encoded by POSTN gene) and versican (encoded by VCAN gene), as well as a tissue inhibitor of metalloproteinase (encoded by TIMP1 gene) were also significantly increased in the metastatic site compared with the metastasis-free liver. In contrast, qHSC signatures such as lipin 2 (encoded by LIPIN2 gene) and retinoid X receptor α (encoded by RXRA gene) were downregulated by fourfold after the metastasis.

Next, an aggressive murine CRC liver metastasis model was established via hemi-splenic injection (Fig. 1g) of a highly metastatic subtype of CT26 cells (CT26-FL3)[24]. Consistent with the clinical result, activation of HSCs characterized by α-SMA expression, the single most reliable marker of aHSCs[25], within the metastatic site was observed 13 days post inoculation (Fig. 1b), when macroscopic metastatic lesions appeared. RXFP1 is a leucine-rich G-protein-coupled receptor capable of stimulating the NO/cyclic guanosine monophosphate (cGMP) pathway, which could inhibit TGF-β/Smad2/3-mediated fibrogenesis[26]. Interestingly, it was significantly upregulated in the metastatic liver and preferentially induced on aHSCs, but not introduced by the CT26-FL3 cells (Fig. 1b, c). Enlightened by this observation, we hypothesized that there might be an endogenous regulatory mechanism against the overwhelming fibrosis during the liver injury. To prove that, methionine-choline-deficient (MCD) diet-induced nonalcoholic steatohepatitis (NASH) model and carbon tetrachloride (CCl4)-induced liver fibrosis, two classic experimental murine liver fibrosis models, were used to study the

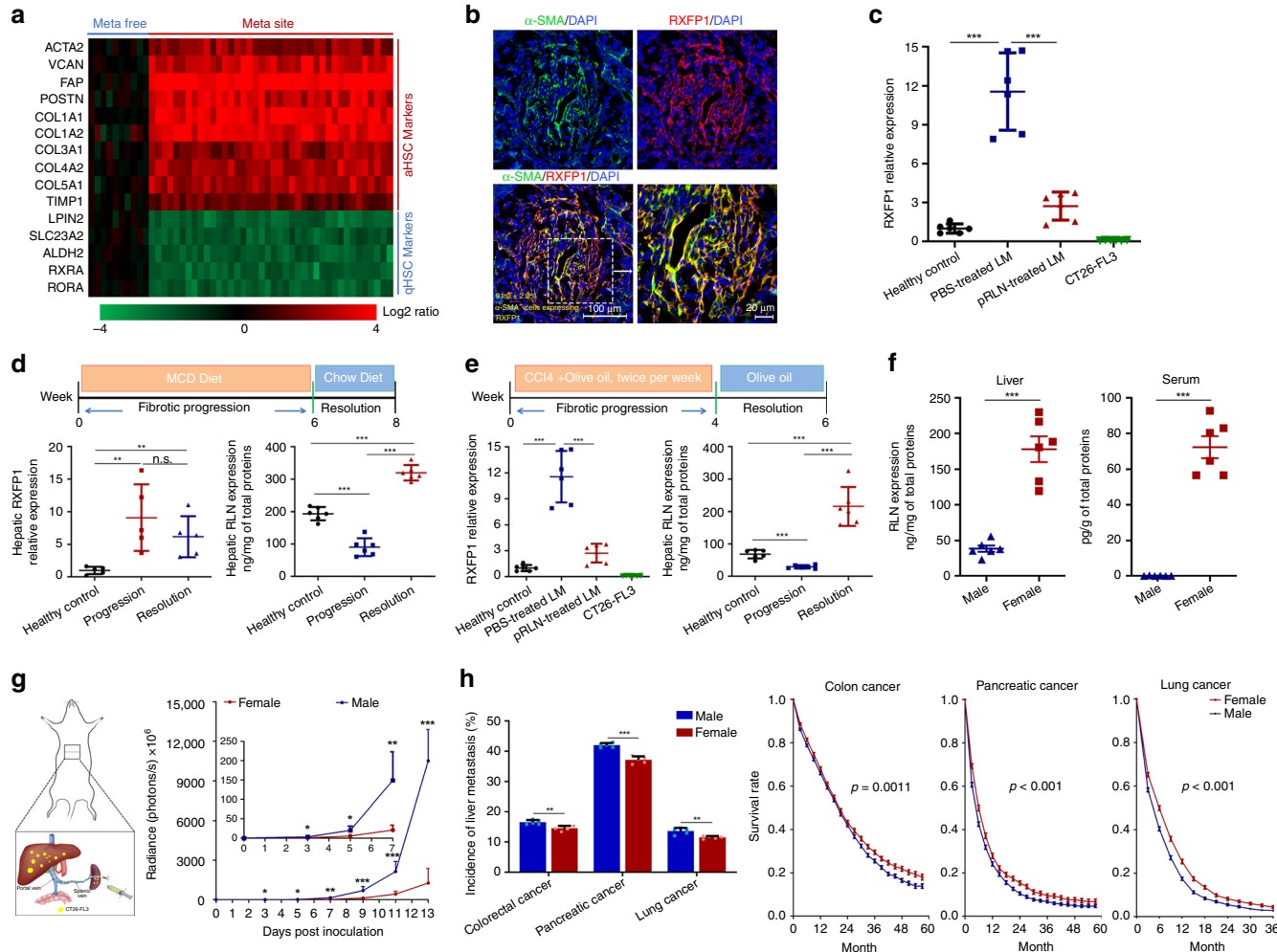

**Fig. 1** HSC activation in liver metastasis and an endogenous regulation mechanism. **a** Heat map comparing activated and quiescent HSC (aHSC and qHSC) gene signatures between metastatic-free sites and metastatic sites in CRC liver metastasis patients. Columns represent 50 GEO patient samples grouped according to different collection sites (metastasis-free (meta-free) and metastatic site (meta site)); rows represent HSC markers sorted by different states (activation and quiescence). Values represent the log2 ratio over control (gene expression in metastasis-free samples). **b** Expression of RXFP1 on aHSCs in the CT26-FL3 metastatic liver. Numbers in yellow indicate the average % of α-SMA-positive cells expressing RXFP1 quantified in five randomly selected fields per mouse ($n = 3$). Bars in lower left and right panels represent 100 and 20 μm, respectively. **c** Relative RXFP1 expression in CT26-FL3 metastatic livers of PBS and pRLN LCP-treated groups as well as in CT26-FL3 cell lines were compared with healthy livers by quantitative RT-PCR ($n = 6$). LM liver metastasis. **d**, **e** Hepatic RXFP1 and RLN expression at different stages of MCD diet-induced NASH model (**d**) or CCl4-induced liver fibrosis (**e**). Relative RXFP1 mRNA and RLN peptide expressions in the fibrotic livers at different stages were compared with healthy livers by quantitative RT-PCR ($n = 5$) and ELISA ($n = 6$), separately. **f** RLN peptide expression in the liver and serum of male and female mice bearing CT26-FL3 liver metastasis ($n = 6$). **g** Schematic of CRC liver metastasis model via hemi-splenic injection and gender disparity of the metastatic progression monitored by bioluminescence analysis ($n = 5$). **h** Differences of the incidence and survival of synchronous liver metastases from CRC (4898 males vs. 3460 females), pancreatic cancer (2708 males vs. 1718 females), and lung cancer (3001 males vs. 2256 females) in male and female patients aged 15–60 based on the clinical data (2010–2014) acquired from SEER database. Significant differences were assessed in (**c**–**g**) (each time points), and (**h**) left panel using *t* test, in (**h**) right panels using log rank test. Results are presented as mean (SD). *$p < 0.05$, **$p < 0.01$, ***$p < 0.001$, n.s., not significant

dynamic expression of RXFP1 and its primary endogenous ligand, mouse RLN1 (equivalent to human RLN2, the major stored and circulating form of the RLN family, hereinafter referred as RLN), in the liver at different stages of fibrosis including the healthy state, fibrosis progression, and resolution. The recovery from liver injuries was characterized by reverted blood levels of alanine aminotransferase (ALT) and aspartate transaminase (AST) (Supplementary Fig. 1), two major enzymes indicating liver damages. In both models, hepatic RXFP1 was upregulated by approximately ninefold during continuous liver injuries, which allowed aHSCs to potentially convert back to the quiescent state once binding with RLN (Fig. 1d, e). RLN expression in the liver was suppressed significantly as fibrosis

progressed, which might associate with an activated inhibitory pathway against RLN production during the profibrogenic process. Nevertheless, the hepatic RLN level bumped up by about 9.2-fold comparing with that under continuous liver injuries once the injury factors disappeared. This 2–3-fold higher RLN level than the basal healthy state thereby promoted fibrosis resolution (Fig. 1d, e). As expected, downregulation of RXFP1 was observed during the resolution, which might be attributed to the partial reversion of aHSCs.

Systemic H2-RLN has been reported to be 10–50-fold higher in females than in males[16], which might be contributed by distinct transcriptional regulation via estrogen and androgen[27,28]. Similarly, we observed much higher RLN levels in the serum and

liver from female mice compared to those from male mice of the same age, both bearing CT26-FL3 liver metastasis (Fig. 1f). Moreover, hemi-splenic injection of $1 \times 10^5$ CT26-FL3 cells resulted in more aggressive liver metastasis progression in male mice than in female mice with a significant difference ($p < 0.05$) already detected on day 3 post inoculation (Fig. 1g). Based on the clinical data acquired from the US Surveillance, Epidemiology, and End Results Program (SEER) database, we found female patients had lower incidences of synchronous liver metastases from CRC, pancreatic cancer, and lung cancer, three major cancer types prone to liver metastasis[29], compared to male patients (14.6% vs. 16.5%; 37.2% vs. 42.0%; 11.5% vs. 13.6%, female vs. male). Female patients with these liver metastases also had a significantly better survival during 5-year follow-up (Fig. 1h). These gender-disparity patterns demonstrate an inherent privilege against liver metastasis in females, which might be associated with higher endogenous RLN level in the liver, and suggest a possible therapeutic potential of RLN for the treatment of liver metastases.

**Design of an in situ RLN depot**. LCPs have shown a high loading capacity (~50–60%)[30] for plasmid DNA, and contained an endosomal escape mechanism based on increased osmotic pressure. The high surface PEGylation tolerance and a small particle size, which could direct the particles away from Kupffer cells (KCs)[31], confer LCPs advantages for liver-specific gene delivery. Sig-1R was significantly upregulated in the metastatic liver compared to the healthy control (Fig. 2a). More specifically, Sig-1R primarily located surrounding the RFP-positive CT26-FL3 cells or α-SMA-positive aHSCs instead of overlapping with the cytosolic RFP or α-SMA. Although Sig-1 R is an endoplasmic reticulum (ER) chaperone protein, it can be upregulated and translocated to the plasma membrane under prolonged ER stress such as high proliferation as an adaptive response[32,33], which might lead to the surface expression of Sig-1R on both tumor cells and aHSCs. After identifying Sig-1R as the potential target,

AEAA-conjugated LCPs were employed for the local gene expression in the liver metastasis site (Fig. 2b). The final nanoparticles were about 30 nm in diameter (Supplementary Fig. 2b) with a neutral surface charge (Supplementary Fig. 2c) according to dynamic light scattering analysis. Transmission electron microscopy (TEM) images confirmed the size of LCPs and indicated the spherical shape and homogenous distribution for both CaP core and the final particles (Supplementary Fig. 2a). Using the Cy5 labeled pDNA, the gene encapsulation efficiency was determined to be $52.2 \pm 5.2\%$ ($n = 3$). Hydrophobic dye DiD (0.2 mol %) was incorporated into the outer leaflet lipid of LCPs for the biodistribution study. AEAA-conjugated LCPs predominantly accumulated in the metastatic liver 24 h after i.v. injection into mice bearing CT26-FL3 liver metastasis, whereas nontargeted LCPs showed 2.4-fold less liver-specific distribution (Fig. 2c). No significant difference of liver accumulation was observed between AEAA-conjugated and unconjugated LCPs in healthy mice (Supplementary Fig. 3). The 3.6-fold increase of intracellular uptake of Cy5-pDNA into CT26-FL3 cells by AEAA-conjugated LCPs compared with nontargeted LCPs further confirmed successful targeting of the tumor cells via AEAA (Supplementary Fig. 4).

In vivo expression of the green fluorescent protein plasmid (pGFP) and pRLN in mice with CT26-FL3 liver metastasis was next investigated using confocal microscopy, quantitative real-time PCR (qPCR), and ELISA assay to ensure preferential expression of LCP delivered pDNA within liver metastatic lesions. GFP signal could be detected in the metastatic site on day 1 after the final pGFP LCP injection, which peaked on day 2 and transiently lasted up to 6 days followed by gradual diminishment (Supplementary Fig. 5a). Specifically, GFP was mainly expressed by tumor cells and aHSCs, together occupying more than 85% of GFP-positive cells on day 2. In addition, tumor cells displayed about twofold greater gene expression capacity than aHSCs (Supplementary Fig. 5b). No GFP expression was found in the metastasis-free area in the liver (Supplementary Fig. 6). Significant upregulation of RLN at mRNA

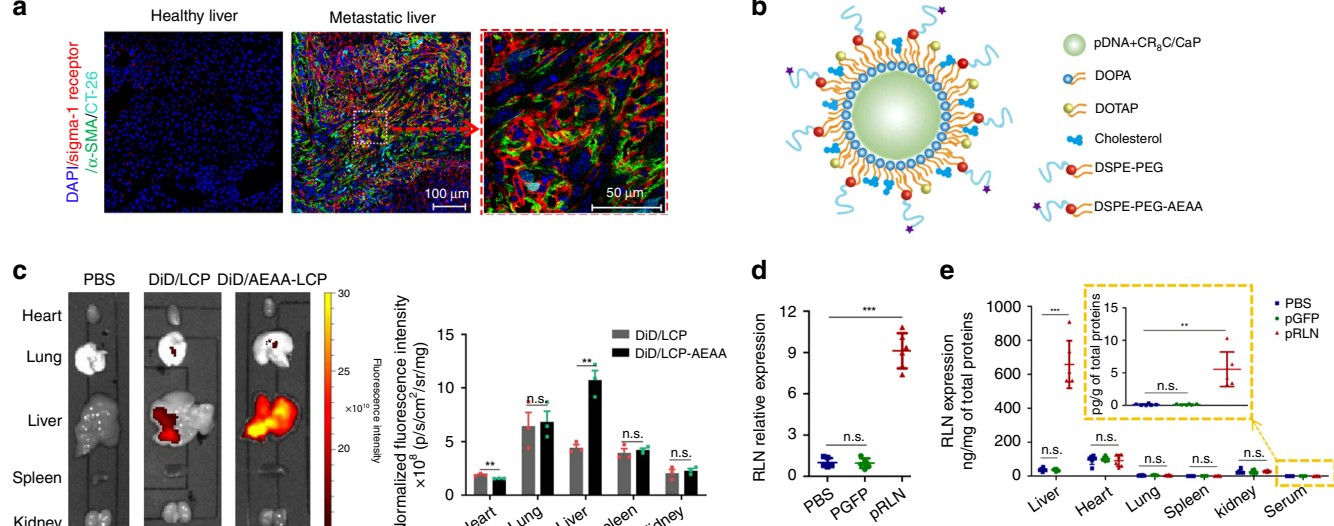

**Fig. 2** Design of locally expressed RLN within the metastatic lesion. **a** Expression of sigma-1 receptor on aHSCs. Immunofluorescence staining of CT26-FL3 metastatic liver from PBS group using anti-α-SMA (green), anti-sigma-1 receptor (red) antibodies, and DAPI (blue). CT26-FL3 cells were colored cyan. Bars in the middle and right panels represent 100 and 50 μm, respectively. **b** Schematic of LCP particles for pRLN delivery. **c** Images and quantitative results of the DiD-loaded LCP (DiD/LCP) or AEAA-conjugated LCP (DiD/AEAA-LCP) in the metastatic liver and other major organs at 24 h after injection in mice bearing CT26-FL3 liver metastasis for 13 days ($n = 3$). **d** Relative RLN mRNA expression in the liver of male mice bearing CT26-FL3 liver metastasis 2 days after the final pDNA LCP injection (30 μg pDNA per mouse every 2 days for totally three injections, $n = 5$). **e** RLN peptide expression in the major organs and serum 2 days after the final pDNA LCP injection ($n = 5$). Significant differences in (**c**–**e**) were assessed using $t$ test. Results are presented as mean (SD). **$p < 0.01$, ***$p < 0.001$, n.s., not significant

and protein levels in the metastatic liver were verified 2 days after three pRLN LCP i.v. injections (Fig. 2d, e). As a secretory peptide, the RLN gene contains an endogenous signal motif that commits the peptide for extracellular section[34]. The close location of aHSC surrounding tumor cells within metastatic lesions facilitates the binding of paracrined RLN with RXFP1 on aHSCs. It should be noticed that RXFP1 and other subtypes of relaxin family peptide receptors (e.g., RXFP2) are widely expressed in other peripheral tissues with different roles[35]. Off-target expression of RLN in other organs can therefore lead to side-effects. Importantly, no significant difference of RLN expression was found in other major organs (Fig. 2e). Increased serum RLN levels (0.2 ± 0.1 pg/mL), which was nevertheless far below the toxicity limit (~13 ng/mL)[36], might be associated with leaking of excessive RLN expressed in the metastatic liver into the systemic circulation. These results demonstrate that the AEAA-LCP vector allows for in situ expression in the liver metastatic lesion with minimal off-target effects.

**RLN gene therapy remodels stromal microenvironment.** We next examined the effect of pRLN LCP to deactivate aHSCs within the metastatic lesion in male mice bearing CT26-FL3 liver

metastasis. α-SMA expression in the metastatic lesion was reduced by 50-fold after pRLN LCP treatments compared with PBS control, along with a shrunk metastatic area (Fig. 3a). The collagen content, which is a major ECM protein, decreased dramatically by pRLN LCP treatments (Fig. 3a). The nitric oxide synthase is an important downstream product of RLN-mediated NO pathway. The production of NO and subsequent cGMP finally inhibits Smad2/3 phosphorylation and disrupts the predominant profibrogenic signal transduction[37]. The western blot analysis not only confirmed the downregulation of α-SMA and collagen I, but also demonstrated 2.4-fold increase of inducible nitric oxide synthase (NOS2) after the treatment (Fig. 3b). As expected, pRLN LCP administration resulted in more than two- and threefold reduction of pSmad2 and pSmad3, respectively, compared to the PBS and pGFP LCP treatment (Fig. 3b). Because of aHSC reversion, the prometastatic chemokine, CXCL12, was also downregulated (Fig. 3b). Furthermore, TGF-β, platelet-derived growth factor (PDGF), and fibroblast growth factor (FGF), which can be secreted by tumor cells, immune cells (e.g., KCs and infiltrating monocytes), and aHSCs[15], were significantly reduced (Fig. 3c). The downregulation of these profibrogenic

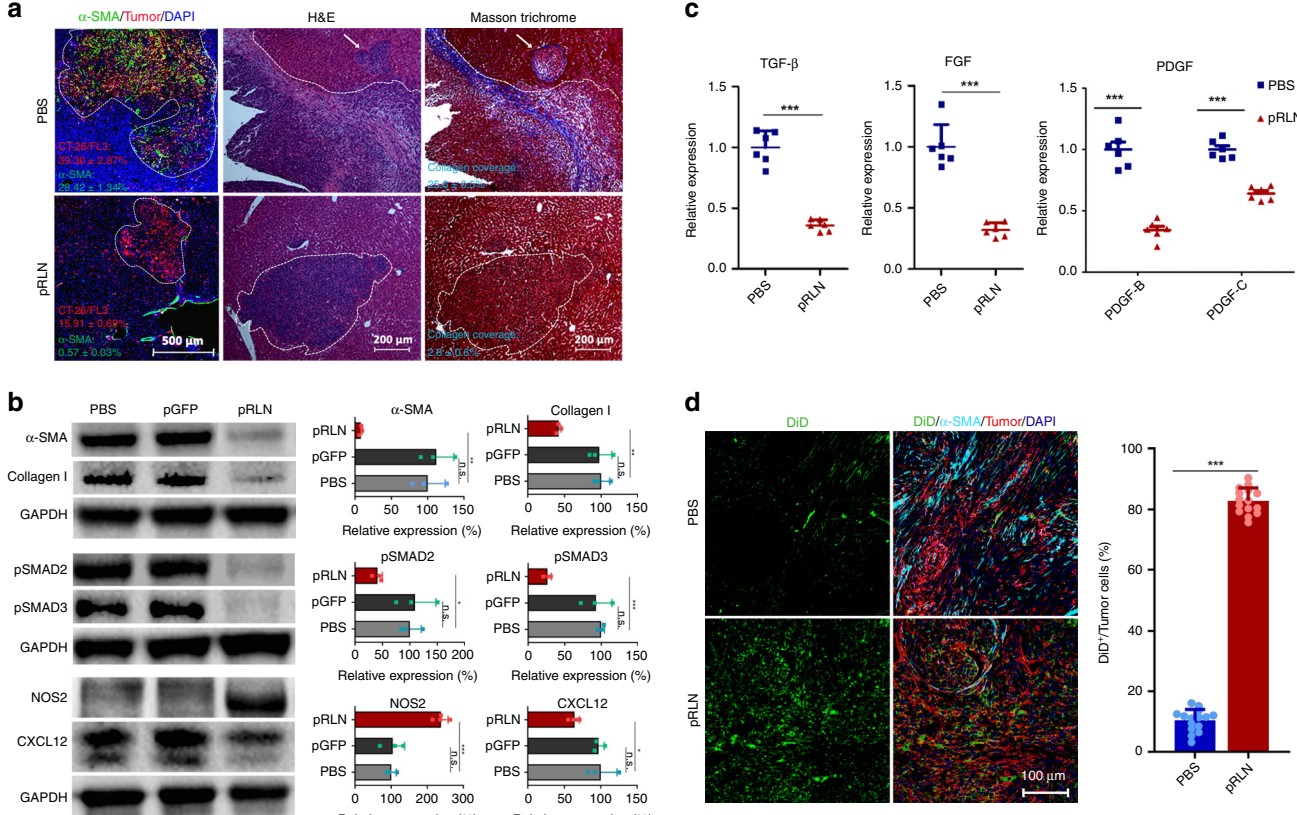

**Fig. 3** RLN gene therapy remodels stromal environment in the metastatic lesion. **a** α-SMA and collagen downregulation and shrunk metastatic foci 4 days after three pRLN LCP (30 μg pDNA every 2 days) i.v. injections in male mice bearing CT26-FL3 liver metastasis. Immunofluorescence staining of CT26-FL3 metastatic liver from PBS and pRLN treatment groups using anti-α-SMA (green) antibody and DAPI (blue). CT26-FL3 cells were colored red. Numbers in corresponding colors indicate the average % of tumor cells and α-SMA-positive cells quantified in five randomly selected fields per mouse (n = 3). H&E and Masson's trichrome staining were performed in adjacent sections. White dotted line and arrow indicate the metastatic foci. Bars in the left two, right four panels represent 500, 200 μm, respectively. The average % collagen (blue staining) coverage was calculated in five randomly selected fields per mouse (n = 3). **b** Western blot analysis of aHSC markers (α-SMA and Collagen I, CXCL12), TGF-β downstream proteins (pSMAD2 and pSMAD3), and NO pathway indispensable protein (NOS2) in CT26-FL3 metastatic livers after RLN gene therapy (n = 3). **c** Relative mRNA expression of profibrogenic factors (TGF-β, FGF, and PDGF) in CT26-FL3 metastatic livers after RLN gene therapy (n = 6). **d** Distribution of second-wave nanoparticles (DiD/LCP) in the metastatic liver of mice receiving PBS or RLN gene therapy. Immunofluorescence staining of CT26-FL3 metastatic liver using anti-α-SMA (cyan) and DAPI (blue). DiD/LCP and tumor cells were colored green and red, respectively. % DiD positive signals are normalized by tumor cell occupation in five randomly selected fields per mouse liver (n = 3). Bar represents 100 μm. Significant differences were assessed using t test. Results are presented as mean (SD). *p < 0.05, **p < 0.01, ***p < 0.001, n.s., not significant

factors were presumably related to the inhibited metastasis, deactivated aHSCs, and modified immune milieu in the metastatic lesion. These data supported that aHSCs in the metastatic liver were shifted from an activated to the quiescent state by the enforced expression of RLN.

Stromal fibroblasts construct the major physical barrier for the nanoparticle-based delivery into solid tumors due to their capacity to secrete massive ECM, which limits the interstitial transport due to high interstitial fluid pressure and interconnected dense network[38], a case also applicable in liver metastases. Therefore, remodeled stromal environment after pRLN LCP treatments was further verified by the penetration of a second-wave nanoparticle administration. A single-dose of DiD-loaded LCPs were i.v. administrated into CT26-FL3 liver metastasis mice 3 days after pretreatments with pRLN LCPs or PBS. An approximately sixfold increase of DiD-labeled LCPs was observed 24 h after the injection in the pRLN LCP-treated liver metastasis compared to the PBS control (Fig. 3d).

**RLN gene therapy synergizes with checkpoint blockade therapy.** We next assessed antimetastasis efficacy of the pRLN LCP treatment on the CT26-FL3 liver metastasis model in both male and female mice. Considering the aggressiveness of the hemi-spleen liver metastasis model, treatments began on day 4 post tumor inoculation, when the bioluminescence already detectable in the liver (Supplementary Fig. 7), according to the scheme illustrated in Supplementary Fig. 8a. No significant difference of metastatic growth was observed between PBS and pGFP LCP-treated mice in both genders (Fig. 4d and Supplementary Fig. 8b). Similar to the gender-disparity profile observed during the liver metastasis progression (Fig. 1g), female mice showed better response to the RLN gene therapy than male mice (Fig. 4b and Supplementary Fig. 8). The lower tumor growth rate in females is presumably related to the 4.7-fold higher hepatic level of endogenous RLN in females than in males (Fig. 1f). In contrast with no significant changes of the tumor growth rate in female mice, decreased tumor growth was observed with the accumulation of pRLN LCP treatments in male mice, which further supports this dose-dependent efficacy (Fig. 4b).

Eight doses of the free RLN peptide was administrated daily as another control (Fig. 4a). The dose of 30 μg/kg was used according to a Phase II trial for the acute heart failure therapy[39]. No metastatic inhibition effect or survival benefit was observed, which is probably due to the extremely short half-life[40] and low accumulation of the exogenous RLN into the metastatic site.

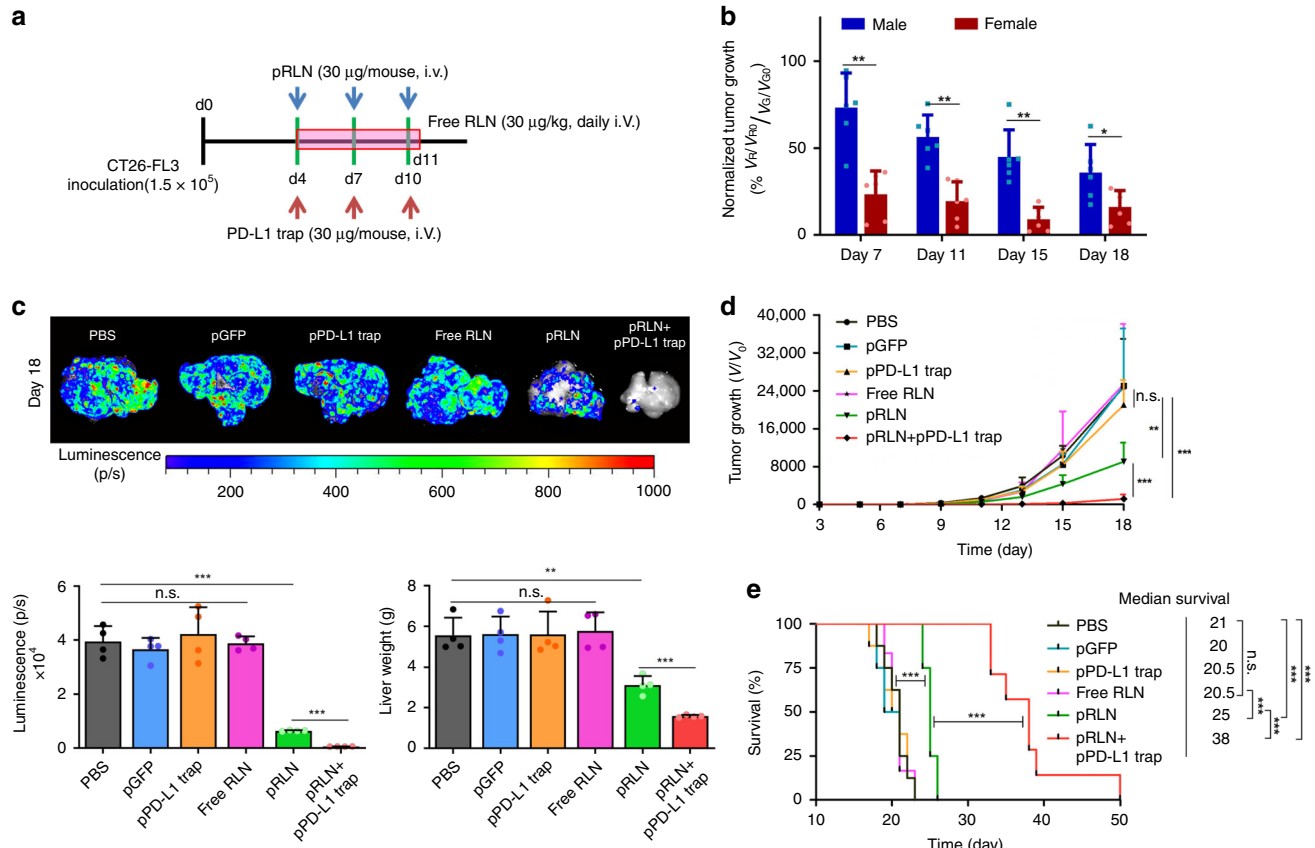

**Fig. 4** RLN gene therapy shows gender-biased response and synergizes with PD-L1 blockade in CT26-FL3 liver metastasis. **a** Tumor inoculation and treatment scheme for pRLN LCP, free RLN, or pRLN + pPD-L1 trap LCP therapy. **b** Difference of normalized tumor growth rate between two genders after pRLN LCP treatment ($n = 6$). The normalized tumor growth rate was calculated comparing the fold-changes over initial of the metastasis volume ($V/V0$) characterized by bioluminescence intensity in the pRLN LCP-treated group with those of the pGFP LCP group. $V_R$, tumor volume after pRLN LCP treatment; $V_{RO}$, initial tumor volume of the pRLN LCP group; $V_G$, tumor volume after pGFP LCP treatment; $V_{G0}$, initial tumor volume of the pGFP LCP group. **c** Imaging and quantification analysis (both bioluminescence and liver weight) of ex vivo CT26-FL3 liver metastasis burden from PBS, pGFP LCP, pPD-L1 trap LCP, free RLN, pRLN LCP, and pRLN + pPD-L1 LCP treated groups 18 days after tumor inoculation ($n = 4$). **d** Metastatic growth curves of CT26-FL3 liver metastasis in each treatment group ($n = 8$). **e** Mice survival curves in each treatment group ($n = 8$). Significant differences were assessed in (**b**) using $t$ test, in (**d**) using two-way ANOVA with multiple comparisons, in (**e**) using log rank test. Results are presented as mean (SD). *$p < 0.05$, **$p < 0.01$, ***$p < 0.001$, n.s., not significant

Due to the constant exposure to non-self-antigens derived from the GI tract, liver has developed immunoregulatory mechanisms to avoid overwhelming inflammation. The widely expressed PD-L1 by multiple liver resident cells including hepatocytes, KCs, liver sinusoidal endothelial cells and aHSCs[41–43], is one of the most important regulatory mechanisms. In the special case of liver metastasis, this immunosuppressive environment can be exploited by metastatic tumor cells to escape T-cell surveillance. Therefore, PD-L1 blockade should be a promising immunotherapy for the treatment of liver metastasis. However, no successful clinical outcomes have been reported for the application of PD-1/PD-L1 blockade immunotherapies in the treatment of liver metastasis. Here, we tested the antimetastatic efficacy of PD-L1 blockade immunotherapy in the CT26-FL3 liver metastasis model through the delivery of a PD-L1 trap plasmid (pPD-L1 trap). The pPD-L1 trap encodes a secretory fusion protein, which contains trimerized extracellular domain of PD-1 that can specifically bind to PD-L1 with high affinity[44]. The AEAA-modified LCP system was also utilized for the targeted delivery of pPD-L1 trap to the metastatic lesions, which allows to avoid systemic toxicity and minimize off-target effects to the normal liver parts[44]. Neither antimetastatic effect nor survival benefit was observed after pPD-L1 trap LCP treatments (Fig. 4a, d, e). Metastatic tumor cells spread all over the liver 18 days after inoculation despite the PD-L1 blockade therapy (Fig. 4c). Nevertheless, the combination of the RLN gene therapy with the PD-L1 blockade (Fig. 4a) produced a synergistic antimetastatic effect in male mice bearing CT26-FL3 liver metastasis (Fig. 4d and Supplementary Fig. 7). While the pRLN LCP treatment alone only increased the median survival by 25%, this combination therapy contributed to almost twofold increase in the median survival compared with the PBS or pGFP group (Fig. 4e). Consistently, mice receiving the combination therapy had the liver weight comparable to healthy mice, which was attributed to more than 60-fold or 10-fold reduction in liver metastasis burden compared with the PBS or the pRLN LCP treated group (Fig. 4c).

**RLN gene therapy modifies immune microenvironment**. The inert response to PD-L1 blockade and synergistic antimetastasis effect by the combination of pRLN and pPD-L1 trap LCPs suggested possible immune modulation by the RLN gene therapy. Mice from different treatment groups were sacrificed 4 days after the last treatment to ensure reliable immune analyses. Microscopic analysis demonstrated aHSCs function as a robust barrier to block the CD3+ T-cell infiltration into metastatic lesions (Supplementary Fig. 9). The pRLN LCP treatment resulted in the reduction of aHSCs and a significant alleviation of lymphocyte exclusion. The pRLN + pPD-L1 trap LCPs further enhanced T-cell accumulation into metastatic lesions. Flow cytometry analysis confirmed the pRLN LCP treatment increased CD8+ and CD4+ T cells by approximately threefold in the metastatic lesion. The pRLN + pPD-L1 trap treatment increased infiltrating CD8+ T cells more significantly than CD4+ T cells, leading to the highest CD8+/CD4+ ratio among all groups (Fig. 5b and Supplementary Fig. 10). The important involvement of CD4+ and CD8+ T cells in the pRLN + pPD-L1 trap treatment was further verified by the depletion study, in which anti-CD4 or anti-CD8 monoclonal antibodies (mAbs) significantly compromised the therapeutic efficacy compared to the IgG control (Fig. 5c). Cytokines and chemokines are important mediators to manipulate the immune microenvironment. Particularly, Th2 cytokine IL-4 has a profibrotic effect through driving the differentiation of quiescent resident fibroblast to myofibroblast[45]. CCL2 and CCL5 exert the major function to recruit myeloid cells. After being

stimulated by the immunosuppressive Th2 cytokines including IL4, IL6, and IL10[12], these recruited myeloid cells will differentiate towards immunosuppressive phenotypes (e.g., immature dendritic cells (DCs), type II macrophages (M2), and MDSCs). The RLN gene therapy substantially downregulated these cytokines as well as CCL2 and CCL5 chemokines in metastatic lesions. In contrast, the pRLN LCP treatment significantly increased the expression of Th1 cytokines such as IFN-γ and IL12 (Fig. 5a), which potentiate cytotoxic T-cell killing and overcome immunosuppressiveness[46]. Accordingly, Treg cells and MDSCs were significantly reduced in both pRLN and pRLN + pPD-L1 trap groups, along with M2 skewing towards the proinflammatory type I macrophage (M1) phenotype (Fig. 5b). The level of activated DCs was only increased by the pRLN + pPD-L1 trap LCP treatment, which might relate with PD-L1-mediated inhibition of DC maturation[47]. Overall, these data suggest the capacity of the RLN gene therapy to switch the immunosuppressive microenvironment within liver metastasis to an immunostimulatory state. The activated immune environment can be further potentiated by the addition of the local PD-L1 blockade.

**Toxicity evaluation**. Acute or chronic toxicities of each treatment were evaluated by blood cell counts and blood chemistry 2 h, 24 h or 7 days after the final injection (Supplementary Fig. 11a, b, and c). Continuous subcutaneous infusion of *serelaxin* has been reported to prolong the bleeding episodes due to the suppression of platelet release from megakaryocytes and impaired platelet aggregation[48,49]. This bleeding tendency will lead to increased risk of mortality in patients undergoing diagnostic or therapeutic invasive procedures[50], and therefore become one of the most significant safety issues for its application in liver metastasis. Nevertheless, neither free RLN nor pRLN LCP treatment caused significant acute reduction of platelet counts compared with the PBS control. A sudden increase of white blood cell (WBC) counts along with significant fluctuations of lymphocyte and neutrophil fractions in WBCs were detected 2 h after the free RLN injection compared with the PBS control (Supplementary Fig. 11a). In contrast, pRLN LCP treatment did not induce WBC changes either 2 h or 24 h after administration, which suggested lower immunogenicity of the nanoparticles than the recombinant peptide. This inflammatory effect of recombinant RLN is mild and transient as evidenced by the disappearance of significant changes in WBC counts and cell fractions 24 h after the treatment (Supplementary Fig. 11b). No abnormal changes of red blood cell (RBC) counts, alanine aminotransferase (ALT), aspartate aminotransferase (AST), creatinine, or blood urea nitrogen (BUN) were observed 24 h after the last injection in each group. The levels of WBC counts, ALT, AST, and BUN in PBS, pGFP LCP, pPD-L1 trap LCP, and free RLN-treated groups dramatically increased above normal ranges 7 days after the final treatment (Supplementary Fig. 11c). These abnormal hematological changes were probably associated with massive liver injuries caused by metastatic tumor growth. The significant upregulation of lymphocyte and downregulation of neutrophil fractions in WBCs by pRLN and pRLN + pPD-L1 trap further supported less expansion of the liver metastasis after these treatments (Supplementary Fig. 11c).

Immune checkpoint inhibitors are known to associate with immune-related adverse events (irAEs) driven by unmasking self-reactive immune cells. IL-17 and Th17 cells are highly upregulated in inflammatory tissues of autoimmune diseases. Hence, the ratio of Th17 cells can be used as a parameter to monitor the irAEs of checkpoint inhibitor immunotherapy. A previous study has confirmed elevated Th17 cells in the spleen after systemic anti-PD-L1 mAb treatment[44]. In contrast, neither

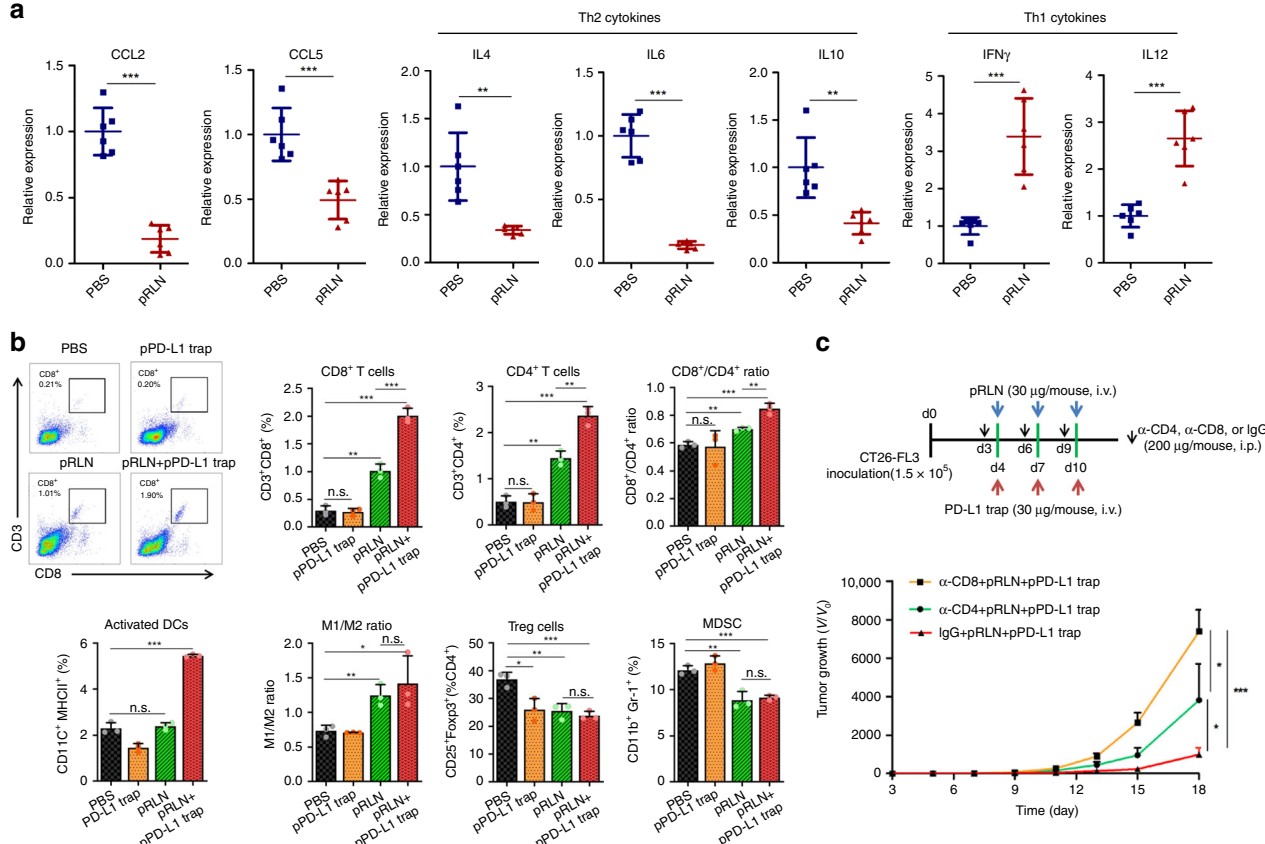

**Fig. 5** pRLN therapy remodels immune microenvironment in the metastatic lesion. **a** Relative mRNA expression of chemokines and cytokines in the liver of male mice bearing CT26-FL3 liver metastasis receiving PBS or pRLN therapy on day 14 (4 days after the last injection) ($n = 6$). **b** CD8$^+$ T cells, CD4$^+$ T cells, CD8$^+$/CD4$^+$ ratio, activated DCs, M1/M2 ratio, Treg cells, and MDSC in the liver of male mice bearing CT26-FL3 liver metastasis receiving various treatments on day 14, analyzed by flow cytometry ($n = 3$). **c** CD4$^+$ or CD8$^+$ T-cell depletion scheme for pRLN + pPD-L1 trap group and the metastatic growth curve ($n = 5$). Significant differences were assessed in (**a**, **b**) using $t$ test, in (**c**) using two-way ANOVA with multiple comparisons. Results are presented as mean (SD). *$p < 0.05$, **$p < 0.01$, ***$p < 0.001$, n.s., not significant

pPD-L1 trap nor pRLN + pPD-L1 trap treatment groups showed any significant increases of Th17 cells in the spleen compared with the PBS control (Supplementary Fig. 12). Lipopolysaccharide (LPS) is known to promote the generation of Th17 cells[51,52]. Flow cytometry analysis confirmed LPS injection increased Th17 cells by 2-fold while the pRLN + pPD-L1 trap treatment did not stimulate the upregulation of Th17 cells compared with the PBS control (Supplementary Fig. 13). Shrunk liver metastatic lesions after pRLN and pRLN + pPD-L1 trap treatments characterized by H&E staining verified the antimetastasis effects. No histological abnormity of heart, lung, spleen, and kidney was observed in any groups (Supplementary Fig. 14).

**Test in other liver metastasis models**. The liver is also the most common metastatic site for pancreatic ductal adenocarcinoma (PDAC), the most common type of pancreatic cancer with high metastatic potential[53]. Recent studies demonstrated highly fibrotic stroma in liver metastases of PDAC and its critical role to sustain metastatic tumor growth[54–56]. KPC, a genetically engineered mouse model with mutations in proto-oncogene K-Ras and tumor-suppressor p53 mutations, is a clinically relevant model of PDAC. Therefore, experimental liver metastasis of a primary tumor cell line generated from KPC mice was used to test the effect of RLN gene therapy and its combination with PD-L1 blockade immunotherapy. As observed in CT26-FL3 liver metastasis, KPC liver metastasis was refractory to PD-L1 blockade therapy, while the combination of pRLN and pPD-L1 trap

efficiently reduced the metastatic burden (Fig. 6a–c). pRLN LCP treatment alone could substantially slow down the progression of metastatic tumor growth during the dosing period. However, the metastasis developed quickly once the treatment stopped, leading to only slightly prolonged survival (Fig. 6d). In contrast, pRLN + pPD-L1 trap treatment induced tumor regression in 50% of the KPC metastasis-bearing mice (Fig. 6b) and significantly extended the median survival in comparison with PBS control and pRLN treatment groups (Fig. 6d).

The effect of pRLN LCP treatment was further tested in the triple-negative breast cancer 4T1 liver metastasis model in female mice. Strikingly, the RLN gene therapy alone could suspend metastatic progression during the treatment period (Fig. 6e and f). The effect was maintained for another week after the final injection, which led to an approximately 50-fold reduction of metastasis burden compared with the PBS control on day 18 (Fig. 6g and h). Remarkably, one out of the five mice treated with pRLN LCPs showed metastasis regression (Fig. 6f). Accordingly, the RLN gene therapy significantly prolonged the median survival by twofold compared with the PBS or pGFP LCP treatment (Fig. 6i).

## Discussion
Stromal remodulation is a growing therapeutic strategy for the treatment of liver metastasis[57–60]. In this work, we discovered an endogenous repair mechanism mediated by RLN/RXFP1 for the resolution of liver fibrosis. In both liver fibrosis models, the

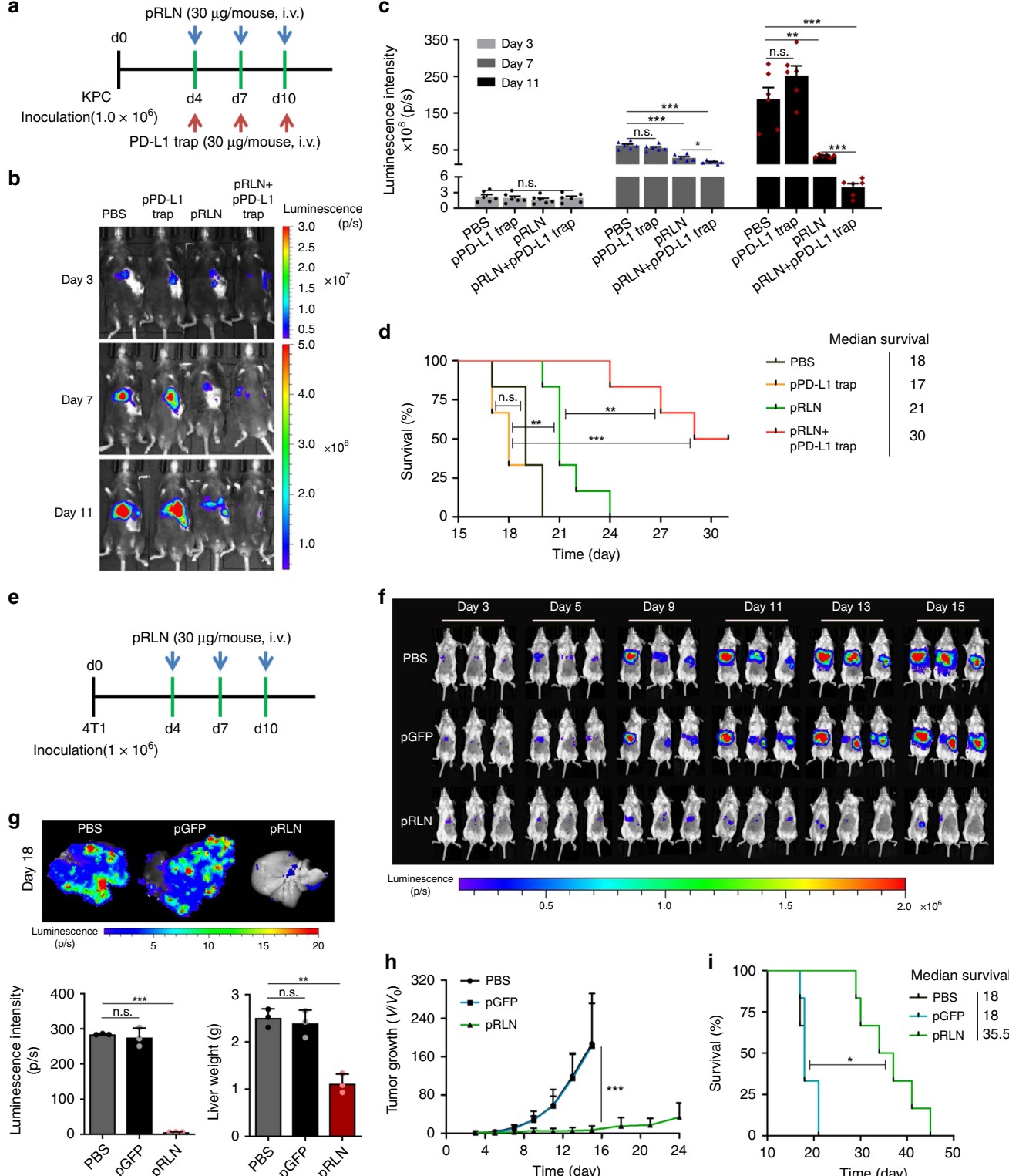

**Fig. 6** RLN and PD-L1 blockade gene-immune therapy on KPC and 4T1 liver metastases. **a** KPC tumor inoculation and treatment scheme. **b** Representative in vivo bioluminescence imaging of mice bearing KPC liver metastasis receiving various treatments on days 3,7, 11 post tumor inoculation. **c** Quantification analysis of KPC liver metastasis burden by bioluminescence intensity ($n = 6$). **d** Survival curves of mice bearing KPC liver metastasis in each treatment group ($n = 6$). **e** 4T1 tumor inoculation and treatment scheme. **f** Representative in vivo bioluminescence imaging of mice bearing 4T1 liver metastasis receiving various treatments. **g** Imaging and quantification analysis (both bioluminescence and liver weight) of ex vivo 4T1 liver metastasis burden from PBS, pGFP LCP, and pRLN LCP 18 days after tumor inoculation ($n = 3$). **h** Metastatic growth curves of 4T1 liver metastasis in each treatment group ($n = 5$). **i** Mice survival curves in each treatment group ($n = 5$). Significant differences were assessed in (**c**, **g**) using $t$ test, in (**h**) using two-way ANOVA with multiple comparisons, in (**d**, **i**) using log rank test. Results are presented as mean (SD). *$p < 0.05$, **$p < 0.01$, ***$p < 0.001$, n.s., not significant

hepatic expression of RXFP1 increased by tenfold under constant liver injuries and decreased gradually during the resolution stage. The concurrent increase of hepatic RLN during the fibrosis resolution suggested its important role as an endogenous fibrosis repair mediator in the liver. On the other hand, the clinically relevant "female privilege" against liver metastases and higher RLN expression in the female liver implies the therapeutic potential of RLN for liver metastasis. As a proof of concept, RLN gene therapy was tested in murine CRC, PDAC, and breast cancer liver metastasis models. AEAA-conjugated LCPs encapsulating pRLN enforced local and transient expression of RLN within metastatic lesions. The pRLN LCP treatment effectively inhibited the growth of CRC liver metastasis in both genders with a better response in female mice. This gender-bias in antimetastasis efficacy is probably associated with the wide gap between endogenous hepatic RLN levels (~5-fold difference) in male and female mice. The difference in therapeutic efficacy can be minimized by gradual accumulation of exogenous RLN via pRLN LCP treatments. On the other side, RLN receptor isoforms (including RXFP1) are regulated by sex-steroid as well. Progesterone and estrogen upregulate while testosterone and androgen downregulate RXFP1 expression[61], which may lead to the worse response towards the RLN gene therapy in male mice. Substantially modulated stromal environment was confirmed by aHSC deactivation, ECM degradation, and downregulation of profibrogenic factors. Recent studies on PD-L1 checkpoint-based immunotherapy have demonstrated the importance of intratumoral T-cell infiltration to achieve the optimum antitumor efficacy by PD-L1 blockade[44,62]. Unfortunately, the stromal barrier and immunosuppressive milieu limited penetration of T cells into the metastatic lesion, which led to unresponsiveness to PD-L1 trap immunotherapy. pRLN LCPs successfully upregulated intrametastasis CD4+ and CD8+ T cells and downregulated immunosuppressive Treg and MDSCs, along with switched phenotypes toward immunostimulatory ones for both cytokines and macrophages. The combination of pRLN LCP with PD-L1 trap further enhanced T-cell-mediated surveillance against CRC liver metastasis and significantly prolonged survival by twofold. Notably, the experimental models used in the current study mimic the late stage of liver metastasis due to rapid tumor formation and higher metastatic rate compared with spontaneous metastasis models[63,64], and therefore, these models are well adapted for the evaluation of antimetastasis efficacy of aHSC deactivation strategy in the established and advanced liver metastasis. However, the experimental liver metastasis models bypass early stages of liver metastasis, which possibly limit more accurate evaluation of our therapeutic modality for the reference of future clinical study. It is also noteworthy that despite the positive effect of the RLN gene therapy reported here, previous studies on the prostate cancer demonstrated that tumor cells overexpressing H2-RLN had a greater xenograft tumor volume than wild-type controls due to increased angiogenesis[65]. Mutated H2-RLN, which functioned as an RXFP1 antagonist, impaired prostate tumor growth[66]. Nevertheless, the usage of immune-deficient mouse models in these studies might overlook RLN-related immune involvement, which has been proven crucial for the RLN gene therapy here. Furthermore, in contrast with high RXFP1 expression on the prostate cancer cells in these studies, the lack of RXFP1 expression in both CRC and breast cancer cells (Supplementary Fig. 15) resulted in unresponsiveness of tumor cells to RLN.

In contrast with pRLN LCP treatment, daily systemic administration of RLN peptide did not inhibit the growth of CRC liver metastasis, which highlights the importance of local deposition for the peptide-based treatment to take effect. The AEAA-conjugated LCP vector achieved high intra-metastasis accumulation with the minimal off-target effects. No adverse hematological response or histological toxicity was observed after repeated injections.

In summary, based on an endogenous antiliver fibrosis mechanism, we developed an in situ RLN gene therapy capable of inhibiting the progression of multiple types of liver metastases with high efficacy, safety and low cost. Its potential to synergize PD-L1 blockade immunotherapy for the treatment of liver metastases further increased its translatability to the clinic.

## Methods

**Materials.** N-(Methoxypolyethylene oxycarbonyl)-1,2-distearoryl-sn-glycero-3-phosphoethanolamine (DSPE-PEG) and 1,2-distearoyl-sn-glycero-3-phosphoethanolamine-N-[amino (polyethylene glycol)-2000] (DSPE-PEG-NH2) were purchased from NOF Corporation (Tokyo, Japan). N-(2-aminoethyl)-4-methoxybenzamide-conjugated DSPE-PEG (DSPE-PEG-AEAA) was synthesized according to a previously established protocol[67]. Briefly, 4-methoxybenzoyl chloride and 2-bromoethylamine hydrobromide were mixed at room temperature for 6 h. Then, DSPE-PEG-NH2 was added into the above solvent and stirred in an oil bath at 65–70 °C for 24 h. The final product was precipitated by ether, washed and lyophilized for further use. All other lipids were purchased from Avanti Polar Lipids, Inc. (Alabaster, AL). Peptides (mcCR8C) were purchased from Elim Biopharmaceuticals, Inc. (Hayward, CA); monocyclic abbreviated to mc. DiD was purchased from ThermoFisher Scientific. Methionine and choline-deficient L-amino acid (MCD) diet was purchased from Research Diets, INC (New Brunswick, NJ). Plasmid encoding mouse relaxin 1 (RLN) with Flag tag driven by the cytomegalovirus (CMV) promoter was prepared by Sino Biological Inc (Beijing, China). Plasmid encoding green fluorescence protein (GFP) driven by the CMV promoter was purchased from Bayou Biolabs (Harahan, LA). Plasmid encoding for PD-L1 trap was prepared by Dr. Rihe Liu's lab with the detailed map and DNA sequence first reported by Miao et al.[68]. Recombinant mouse relaxin 1 peptide was purchased from LifeSpan BioSciences, Inc. (Seattle WA). All other chemicals were obtained from Sigma-Aldrich (St. Louis, MO) unless specifically mentioned.

**Cell lines.** Murine colorectal cancer CT26-FL3 cells stably expressing red fluorescent protein (RFP)/ firefly luciferase (Luc) were established by transfection of CT26-FL3 cells with vectors carrying RFP/Luc and puromycin resistance gene. The original CT26-FL3 cells were kindly provided by Dr. Maria Pena at the University of South Carolina. The primary pancreatic tumor cell line derived from the spontaneous KPC model of PDAC (LSL-Kras G12D/+; LSL-Trp53R172H/+; Pdx1-Cre, of C57BL/6 background) was generously provided by Dr. Serguei Kozlov from the Center for Advanced Preclinical Research, Frederick National Laboratory for Cancer Research (National Cancer Institute). The primary cell line was stably transfected with lentivirus vector with carrying GFP/Luc and puromycin resistance gene. Murine breast cancer 4T1 cells were obtained from Tissue Culture Facility—UNC Lineberger Comprehensive Cancer Center, and further transfected with lentivirus vector carrying GFP/Luc and puromycin resistance gene. CT26-FL3 cells were cultivated in Dulbecco's modified Eagle's medium (DMEM, high glucose, Gibco) supplemented with 10% fetal bovine serum (FBS, Gibco), 1% antibiotic −antimycotic (Gibco) and 1 μg/ml puromycin (ThermoFisher) at 37 °C and 5% $CO_2$ in a humidified atmosphere. KPC cells were cultured in DMEM:Nutrient Mixture F-12 (DMEM/F12), supplemented with 10% FBS, 1% antibiotic—antimycotic and 1 μg/mL puromycin at 37 °C and 5% $CO_2$ in a humidified atmosphere. 4T1 cells were cultivated in RPMI 1640 medium supplemented with 10% FBS, 1% antibiotic−antimycotic and 1 μg/mL puromycin at 37 °C and 5% $CO_2$ in a humidified atmosphere.

**Mouse model establishment.** Eight-week-old male or female BALB/c and C57BL/6 mice were obtained from the Jackson Laboratory. All animal testing and research were conducted in compliance with ethical regulations approved by the University of North Carolina at Chapel Hill's Institutional Animal Care and Use Committee. For CCl4-induced liver fibrosis model[69], male C57BL6 mice received intraperitoneal injection of CCl4/olive oil (7/1, v/v) mixture at a dose of 1 μL/g twice weekly for 6 weeks (mimicking the fibrosis progression stage) or 4 weeks plus olive oil injection for another 2 weeks (mimicking the resolution stage). For MCD-induced NASH model[70], female C57BL6 mice were fed with MCD diet for 8 weeks (mimicking the fibrosis progression stage) or 6 weeks plus normal chow diet for another 2 weeks (mimicking the resolution stage). CT26-FL3 liver metastasis model was established in male or female BALB/c mice. Sub-confluent cells were harvested and washed in phosphate buffered saline (PBS) just prior to spleen implantation. Mice were anesthetized by 2.5% isoflurane and placed in supine position. For splenic inoculation, an incision located below the left rib cage was made to exteriorize the spleen. The spleen was tied and cut into two parts each containing intact vascular pedicle for each half of the spleen. The distal section of the spleen was inoculated with $1.5 \times 10^5$ CT26-FL3 cells in 150 μL PBS. The hemi-spleen containing inoculated cells was resected 5 min after inoculation allowing the cancer cells to enter the portal vein. The hemi-half containing inoculated cells was resected to model primary tumor resection. The other half of the spleen was returned to the cavity and the abdominal wall and skin were closed with 4–0 polyglycolic acid

sutures. Similarly, KPC liver metastasis model was established by hemi-splenic injection of $1 \times 10^6$ KPC cells in 150 μL PBS into male C57BL/6 mice. And 4T1 liver metastasis models was established by hemi-splenic injection of $1 \times 10^6$ 4T1 cells in 150 μL PBS in female BALB/c mice.

**Luciferase imaging of whole animal and ex vivo tissues.** The in vivo metastatic progression was monitored by intraperitoneal injection of 100 μL of D-luciferin (Perkin Elmer, 20 mg/mL) followed by bioluminescent analysis using an IVIS® Kinetics Optical System (Perkin Elmer, CA). For ex vivo imaging, livers after dissection were quickly rinsed three times in PBS and placed in 1 mL luciferin solution (20 mg/mL) for 1 min, followed by immediate IVIS imaging.

**Metastatic growth inhibition assay.** Mice bearing liver metastases were randomized blindly into different treatment groups, and the investigator was blinded to the group allocation during the animal experiments. For outcome assessment, same protocol was applied across experimental groups. PBS, pGFP LCP (30 μg pDNA per mouse, i.v.), pRLN LCP (30 μg pDNA per mouse, i.v.), pPD-L1 trap LCP (30 μg pDNA per mouse, i.v.), pRLN + pPD-L1 LCP (Combo, 30 μg pDNA for each plasmid per mouse, i.v.), recombinant RLN (30 μg/kg[39], i.v.), Combo + anti-mouse CD8α (α-CD8, Bioxcell, clone 53-6.72, 200 μg per mouse, i.p.), Combo + anti-mouse CD4 (α-CD4, Bioxcell, clone GK1.5, 200 μg per mouse, i.p.), and Combo + polyclonal rat IgG (Bioxcell, 200 μg per mouse, i.p.) were given at respective schedules. In vivo metastatic growth was monitored using IVIS system every other day. The increase of tumor volumes was calculated as luminescence intensities (photon/sec) over the initial. At designed time points, 3–5 mice in each group were sacrificed. Metastatic livers were harvested for immunofluorescence staining, Masson's trichrome staining, flow cytometry analysis, ELISA, western blot, and qPCR assays. For survival studies, the final endpoint achieved when one of the following conditions applied: drastic weight gain or loss greater than 10% within 1 week, or clear signs of distress were detected, such as dehydration, inactivity, lethargy.

**Immunofluorescence (IF) and histology staining.** IF staining was performed on frozen sections of metastatic livers. Tissues for frozen sections were resected and rinsed in PBS, and placed in 4% paraformaldehyde (PFA) overnight at 4 °C. Following 4% PFA fixation, tissues were dehydrated with 15% and 30% sucrose solution overnight at 4 °C. Tissues were snap frozen in O.C.T. (Fisher Scientific, Pittsburgh, PA). Frozen sections were processed through permeabilization and blocking in 5% goat serum at room temperature for 1 h. Primary antibodies were incubated at 4 °C overnight, and fluorescent secondary antibodies were incubated at 37 °C for 1 h. Finally, the slices were mounted with Prolong® Diamond Antifade Mountant with DAPI (ThermoFisher Scientific). Primary antibodies, fluorescent primary and secondary antibodies used for immunofluorescence staining were listed in Supplementary Table 1. IF images were taken on a laser scanning confocal microscope (Zeiss LSM 700). Quantification of positive signals was performed with Image J software. Histology staining was performed on paraffin-embedded sections from metastatic livers and other major organs. All paraffin-embedded tissue was resected, rinsed in PBS, and placed in 4% PFA for 48 h at 4 °C. Following 4% PFA, tissues were rinsed in water and placed in 70% ethanol solution until paraffin-embedded. Masson's trichrome staining were performed following manufacture's protocols (Abcam, Cambridge, MA). Histological images were taken on a microscope (Nikon Eclipse Ti-U).

**Flow cytometry assay.** Metastatic livers were made into single-cell suspensions, and metastasis-infiltrating lymphocytes were quantitatively analyzed by flow cytometry after immunofluorescence staining. In brief, tissues were harvested and digested with collagenase A and DNAase at 37 °C for 45 min. After red blood cell lysis via addition of ACK buffer, cells were collected and dispersed with 1 mL of PBS, and stained by the addition of a cocktail of fluorescence conjugated antibodies. Following staining, cells were fixed with 4% PFA and analyzed via FACS (BD LSR II). Fluorescence conjugated antibodies used for flow cytometry are listed in Supplementary Table 1.

**Quantitative real-time PCR (qPCR) assay.** Total RNA was extracted from the metastatic livers using an RNeasy® Microarray Tissue Mini Kit (Qiagen). cDNA was reverse-transcribed using the iScript™ cDNA Synthesis Kit (Bio-Rad). One hundred and fifty nanograms of cDNA was amplified with the TaqMan™ Gene Expression Master Mix. All mouse-specific primers for RT-PCR reactions are listed in Supplementary Table 2. GAPDH was used as the endogenous control. Reactions were conducted using the 7500 Real-Time PCR System and the data were analyzed with the 7500 Software.

**Western blot and ELISA analysis.** Tissue samples were lysed by a radio-immunoprecipitation assay (RIPA) buffer. A bicinchoninic acid assay (Invitrogen, CA) was used to determine the protein concentration in the tumors according to the manufacturer's protocols. ELISA assay was performed according to the manufacturer's protocols after appropriate titration. For western blot analysis, the protein solution was diluted by sample buffer (4×) containing reducing reagent and

heated at 95 °C for 5 min. Protein was separated by 4–12% SDS-PAGE electrophoresis (Invitrogen), and then transferred to polyvinylidene difluoride membranes (Bio-Rad). The membranes were blocked with 3% bovine serum albumin at room temperature for 1 h and then incubated with primary antibodies overnight at 4 °C. The membranes were washed and further incubated with a secondary antibody (appropriate diluted) at room temperature for 1 h, and then detected using the Pierce ECL Western Blotting Substrate (ThermoFisher Scientific, IL). GAPDH was used as the control.

**Preparation and characterization of LCP loaded with pDNA.** LCP was prepared using a modified protocol[30]. Two separate microemulsions (60 mL each) were prepared of Igepal 520 and cyclohexane (3:7 v/v) and placed under stirring. A pDNA (800 μg, 1 mg/mL) solution was prepared and added into 1800 μL of 2.5 M CaCl$_2$ solution. To this solution, octaarginine peptide (mc-CR8C) was added at an N:P ratio of 2:1 (~820 μg) with well mixture and immediately added to the microemulsion. An $(NH_4)_2HPO_4$ solution (1800 μL, 50 mM) was also prepared and added to the other microemulsion. Each microemulsion was allowed to stir for 20 min. The microemulsion containing $(NH_4)_2HPO_4$ was added to the microemulsion containing the DNA/Peptide/CaCl$_2$. This solution was allowed to stir for 5 min before addition of 1400 μL of 20 mM 1,2-dioleoyl-sn-glycero-3-phosphate (DOPA, in CHCl$_3$). After addition of DOPA the microemulsion was left to stir an additional 30 min. An equal volume of 100% EtOH (120 mL) was added to disrupt the emulsion. The mixture was centrifuged at $10,000 \times g$ for 20 min. After decanting the supernatant, the precipitate was washed twice thereafter with 100% EtOH to remove traces of Igepal and/or cyclohexane. The precipitate was then dried under N$_2$, and resuspended in CH$_2$Cl$_2$. This solution was centrifuged at $13,000 \times g$ for 10 min for the removal of large aggregates, and the supernatant containing the LCP cores (DNA and peptide entrapped within a calcium phosphate nanoprecipitate, supporting and surrounded by a lipid monolayer of DOPA) was recovered.

Final AEAA-LCP-pDNA/mc-CR8C was produced through desiccation of a mixture of free lipids and cores and rehydration via 5% aqueous sucrose solution. The ratio of cores to outer leaflet lipids for optimal final particle formulation was found to be 15 mg core: 420 μL 1,2-dioleoyl-3-trimethylammonium-propane (DOTAP, 40 mM):420 μL Cholesterol (40 mM):666 μL DSPE-PEG2000 (20 mM). Therein, 35 mol% DOTAP, 35 mol% cholesterol, and 30 mol% DSPE-PEG2000 (or 25 mol% DSPE-PEG and 5 mol% DSPE-PEG-AEAA) were utilized as outer leaflet lipids. Zeta potential and particle size of LCP were measured using a Malvern ZetaSizer Nano Series (Westborough, MA). TEM images of LCP were acquired using a JEOL 100CX II TEM (JEOL, Japan)

To characterize pDNA entrapment efficiency of LCPs, pRLN was labeled with Cy5 (Mirus LabelIT kit, Mirus Bio, Madison, WI) according to the manufacturer's instructions. Such Cy5-pDNA was formulated into LCPs. The final nanoparticles were dissolved in the same amount of lysis buffer (2 mM EDTA and 0.05% Triton X-100 in pH 7.8 Tris buffer) at 65 °C for 10 min to release the entrapped pDNA[71]. The standard Cy5-pDNA solutions were prepared by diluting the original Cy5-pDNA in the same lysis buffer. Then, 100 μl of the standard and LCP solution was taken to the 96-well plate and the fluorescence intensity was measured on SpectraMax M5 plate reader (Molecular Devices) with a 620 nm excitation filter and a 670 nm emission filter.

**Biodistribution of LCP nanoparticles.** Approximately 0.2 mol% of hydrophobic dye DiD was incorporated into outer leaflet lipids to formulate the DiD-labeled LCP nanoparticles. After 24 h of intravenous injection of the DiD-labeled LCP nanoparticles, mice were killed and metastatic livers as well as other major organs were collected. The distribution of LCP nanoparticles in the organs was quantitatively visualized with IVIS system, with the excitation wavelength at 640 nm and the emission wavelength at 670 nm.

**In vitro uptake study.** Six-well plates were seeded with $2 \times 10^5$ CT26-FL3 cells per well and incubated in the cell incubator overnight. Different groups of LCP nanoparticles equivalent to 0.5 μg of Cy5-pRLN was added to each well in the presence of DMEM medium and kept in the incubator for 4 h. After incubation, the adherent cells were washed with PBS and detached with 0.05% trypsin. The cells were collected upon centrifugation at $100 \times g$ for 5 min, washed with PBS twice, and resuspended with 400 μL of PBS for immediate FACS analysis.

**In vivo gene expression tracking.** In total three doses of AEAA-conjugated LCPs containing pGFP or pRLN (30 μg pDNA per mouse, $n = 3$ for each group) were every-two-day i.v. injected into 8-week-old BALB/c male mice hemi-spleen inoculated with CT26-FL3 for 4 days. Mice injected with pGFP LCPs were sacrificed 1, 2, 4, 6, and 8 days after the final injection and the metastatic livers were collected and performed with frozen sections, which were further stained with α-SMA fluorescence-conjugated antibody and DAPI. Stained frozen sections were analyzed for cellular GFP expression using confocal microscope. Quantification of the positive signal was performed using Image J software. pRLN LCPs-injected mice were sacrificed 2 days after the final injection. Quantification of RLN expression in the metastatic liver, heart, lung, spleen, kidney, and serum was performed using a mouse relaxin 1 ELISA kit.

**Blood chemistry analysis**. On day 10, 11, or 18 (2 h, 1 day, or 1 week after the final treatments), three mice in each group were subjected to an acute or chronic toxicity assay. Both whole blood and serum were collected. Whole blood cellular components were counted and compared. Creatinine (CRE), blood urea nitrogen (BUN), serum aspartate aminotransferase (AST), and alanine aminotransferase (ALT) in the serum were assayed as indicators of renal and liver functions. Organs including the heart, liver, spleen, lungs, and kidneys were collected and fixed for hematoxylin and eosin (H&E) staining at UNC histology facility to evaluate the organ-specific toxicity and spontaneous metastasis.

**Statistical analysis**. Data were expressed as the mean ± standard deviation (SD). Statistical analysis was performed by the Students' t test when only two value sets were compared. Ordinary two-way ANOVA with multiple comparisons adjusted by Šidák correction was used for comparison between multiple groups. For survival analyses, log rank test was used for comparison. *, **, *** denotes p < 0.05, 0.01, and 0.001, respectively, and was considered significant and documented in figure or figure legend. No exclusion criteria were incorporated in the design of the experiments for this study.

**Reporting summary**. Further information on research design is available in the Nature Research Reporting Summary linked to this article.

### Data availability

The source data underlying Fig. 3b are provided as a Source Data file. All the other data supporting the findings of this study are available within the article and its supplementary information files and from the corresponding author upon reasonable request. A reporting summary for this article is available as a Supplementary Information file.

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

## Acknowledgements
The work in this lab was supported by NIH grants DK100664 and CA198999, and by a grant from Eshelman Institute for Innovation.

## Author contributions
M.H. and L.H. conceived and designed the research. M.H., Y.W., X.Z. performed the in vivo mouse experiments. M.H., Y.W., and S.A. prepared frozen sections, immuno-fluorescence and histological staining. M.H. and Y.W. analyzed and quantified micro-scopic images. M.H., Y.W., and L.X. performed the flow cytometry assay and analyzed the results. M.H., Y.W., and Y.T. analyzed clinical data. J.L. and R.L. designed PD-L1 trap plasmid. M.H. performed the statistical analysis. M.H. and L.H. analyzed the data and wrote the manuscript.

## Additional information

**Competing interests:** The authors declare no competing interests.

