## [Peer Review File · Nature Communications]

Reviewers' comments:

Reviewer #1 (Remarks to the Author):

In this manuscript titled "Fetching an Endogenous Fibrosis Resolution Mechanism: Gene therapy for Liver Metastasis", Mengying and colleagues show that Relaxin, an anti-fibrotic peptide, and its cognate receptor are upregulated in the liver during fibrosis leading to deactivation of HSCs and fibrosis resolution. In an effort to alleviate fibrosis that facilitates liver metastasis, the authors generated nanoparticles to specifically deliver relaxin into experimental liver metastasis. They demonstrate that RLN nanoparticles induce deactivation of the stromal microenvironment and decreased metastatic load. They used two experimental liver metastasis models with breast and colorectal cancer cell lines, and demonstrated that RLN delivery into tumors inhibited metastases formation and prolong survival. Moreover, combination of relaxin gene therapy with anti-PDL-1 immunotherapy had a synergistic effect on metastases inhibition, which they suggested to be induced by immune modulation by RLN, towards an immunosuppressed microenvironment.

The study is original, has novel findings, very interesting and potentially clinically relevant. The manuscript is well written and the data is overall of high quality. However, prior to acceptance several issues should be addressed:

Major points:

1. The authors use models of experimental metastasis rather than spontaneous metastasis, therefore precluding investigation of changes that occur during the early stages of metastasis and the formation of a pre-metastatic niche. While they cannot be expected to change the experimental setting of their study at this stage, this limitation should be clearly stated (by using the term "experimental metastasis") and discussed. Moreover, overstatements like "We evaluated the RLN gene therapy in murine CRC and breast cancer liver metastasis models via hemi-splenic inoculation of CT26-FL3 and 4T1 cells, respectively, which mirrors the clinical scenario of liver metastasis patients.." (introduction section, line 88) should be toned down.
2. In Figure 4, The authors use "tumor growth" (in 4d) and "metastasis inhibition" (in 4b) interchangeably, which is confusing, since in both cases they measure inhibition of the growth of experimental metastasis. This should be corrected.
3. The authors speculate that the reason for the gender bias in liver metastasis is the higher physiological levels of RLN in females. In that case, treatment with exogenous RLN is not necessarily expected to be less effective in males. Nevertheless, their findings indicate a significant gender bias in inhibition of experimental metastasis. This should be discussed.

4. In Figure 5, the authors use FACS analysis to demonstrate that RLN treatment boosted T cell infiltration into metastatic lesions. Immunostaining of metastatic lesions to show actual infiltration would be better than only FACS analysis, especially since their point is that liver fibrosis forms a physical barrier that prevents infiltration, and RLN treatment alleviates it and enables physical entry of immune cells to a presumably less fibrotic tumor.

Showing infiltration by staining is particularly important because according to the methods section, mice were not heart perfused before the metastases were harvested for analysis.

5. Since fibrosis is a major theme of this manuscript, directly demonstrating fibrosis/resolution by staining of collagen content in the liver metastatic lesions (Masson Trichrome, Sirius Red etc.) would more directly substantiate the claim about changes in fibrosis.

Minor points:

Line 44: There is a Typo (“caner” missing c).

Fig. 2C legend: Dil instead of DiD

Line 246: Did the authors mean Fig. 4c?

Fig. 5b: The Y axis title of the graph that shows M1/M2 ratio should be corrected.

Line 300: Should be Fig. 6b and not 6a.

Reviewer #2 (Remarks to the Author):

English has some imperfection and needs to be improved. The number of acronyms is grossly excessive affecting clarity.

The innovation statement in line 26 and 27 “Here, we discovered an anti-fibrotic peptide, relaxin (RLN...” is contradicted by earlier publications, including Clin Med Res. 2005 Nov; 3(4): 241–249, Relaxin: Antifibrotic Properties and Effects in Models of Disease, Chrishan S. Samuel

(see more relevant papers at <http://onlinelibrary.wiley.com/doi/10.1111/bph.v174.10/issuetoc>)

and also earlier discovery of relaxin is subsequently acknowledged in the manuscript. This statement must be modified.

The nanoparticles need to be characterised far more thoroughly and the methods of production were incomplete. It is not clear how much of the gene was actually loaded (we only know how much went into the synthesis). The data in Supplementary Figure 2a were not analysed and the caption or DLS is grossly insufficient and the font size was insufficient.

The nanoparticle production methods are insufficient, in particular they do not articulate how the targeting of nanoparticles was achieved. There is no evidence that targeting is necessary. There are no cell experiments at all, for example confirming successful targeting of cancer (there is only evidence of different liver accumulation which is unconvincing as there is no tumour outline in this figure.. There is no evidence supporting that the nanoparticle design (Fig 2 b) reflects what has been produced.

The nanoparticle toxicity was not fully evaluated. There was no consideration as to the potential immune interactions of the treatment.

Reviewer #3 (Remarks to the Author):

In this manuscript the authors described the role of relaxin (RLN) peptide in liver metastasis. Enforced expression of RLN in liver deactivated the aHSCs and changes in tumor immune microenvironment resulted in controlling the liver metastasis. The authors utilized anisamide conjugated targeted LCP nanoparticles to encapsulate RLN encoded plasmid DNA to target tumor cells and aHSCs. Additionally, nanoparticles containing both RLN and PD-L1 pDNAs prolonged the survival of mice in breast and colorectal cancer liver metastasis models. The subject is interesting and the results are significant. However, the manuscript lacks novelty and the following issues need to be considered prior to publication in nature communications.

Major comments:

1. RXFP is widely expressed in all peripheral tissues of male and female species with different roles in different organs and not restricted to the liver. Additionally, several other RXFP family receptors are expressed in the human body. Interestingly, RLN has high affinity ligand for all RXFP receptors. The authors did not consider these issues and the expression of RLN in other organs might affect their functions? These issues need to be considered and addressed in the manuscript.

2. It is very interesting that anisamide can target HSCs? Are they expressing the sigma receptors? The total expression of protein in the liver could be from tumor cells only? If so how the RLN expression in tumors can control liver metastasis ?
3. Can this therapy be applicable to other cancer metastasis models?
4. If the RLN has significant role in liver metastasis as mentioned in the introduction, why there is no difference in mice's survival between free RLN and pRLN treated groups? Here only combination of pPD-L1 and pRLN showed significant difference in metastasis inhibition. The authors should justify the role of RLN alone in aHSCs deactivation followed by metastasis inhibition.
5. The authors should justify the use of pDNA due to their adverse immune reactions, which is an important aspect in clinical translation.
6. pRLN therapy alone may not be efficient in some tumors (4T1) compared to CT26? Why there was significant difference in therapeutic outcome between the two tumor models with pRLN, where RLN is critical factor in stromal microenvironment? What is the effect of combination therapy on 4T1 metastasis model?
7. Reports also suggesting that RLN expression can increase the risk of prostate cancer and promotes angiogenesis. The authors need to carefully consider and discuss this.

Reviewer reports:

We sincerely apologize for the delay in reaching a decision on your manuscript entitled "Fetching an Endogenous Fibrosis Resolution Mechanism: Gene Therapy for Liver Metastasis". It has now been seen by 3 referees, whose comments are appended below. You will see from their comments copied below that while they find your work of considerable potential interest, they have raised quite substantial concerns that must be addressed. In light of these comments, we cannot accept the manuscript for publication, but would be interested in considering a revised version that addresses these serious concerns.

We hope you will find the referees' comments useful as you decide how to proceed. Should further experimental data or analysis allow you to address these criticisms, we would be happy to look at a substantially revised manuscript. However, please bear in mind that we will be reluctant to approach the referees again in the absence of major revisions. While we expect all reviewer concerns to be addressed comprehensively, we suggest in particular, to confirm T cell infiltration and resolution of fibrosis by IHC/trichome staining, further characterize NPs and their toxicity, explain modest effects with RLN or pRLN alone on survival of mice with liver metastasis. If the revision process takes significantly longer than six months, we will be happy to reconsider your paper at a later date, as long as nothing similar has been accepted for publication at Nature Communications or published elsewhere in the meantime.

We are committed to providing a fair and constructive peer-review process. Do not hesitate to contact us if you wish to discuss the revision or if there are specific requests from the reviewers that you believe are technically impossible or unlikely to yield a meaningful outcome.

When resubmitting your paper, please highlight all changes in the manuscript text file. We also ask that you ensure that your manuscript complies with our editorial policies. Specifically, please ensure that the following requirements are met, and any relevant checklists are completed or updated and uploaded as a Related Manuscript file type with the revised article:

Answer:

Thanks for giving us the opportunity to revise the manuscript. Because we received comments from Refree#1 and Refree#2 earlier on Jan 23, we were able to make the revisions from then on. We really appreciate the early feedback from the editor. We sincerely hope our answers below could address all the concerns, in particular, T cell infiltration and resolution of fibrosis by IHC/trichome staining, further characterize NPs and their toxicity, explain modest effects with RLN or pRLN alone on survival of mice with liver metastasis. We have marked all newly added data and figure numbers in red in both the revised manuscript and responses here.

Referee#1: Cancer Metastasis, therapy

In this manuscript titled "Fetching an Endogenous Fibrosis Resolution Mechanism: Gene therapy for Liver Metastasis", Mengying and colleagues show that Relaxin, an anti-fibrotic peptide, and its cognate receptor are upregulated in the liver during fibrosis leading to deactivation of HSCs and fibrosis resolution. In an effort to alleviate fibrosis that facilitates liver metastasis, the authors generated nanoparticles to specifically deliver relaxin into experimental liver metastasis. They demonstrate that RLN nanoparticles induce deactivation of the stromal microenvironment and decreased metastatic load. They used two experimental liver metastasis models with breast and colorectal cancer cell lines, and demonstrated that RLN delivery into tumors inhibited metastases formation and prolong survival. Moreover, combination of relaxin gene therapy with anti-PDL-1 immunotherapy had a synergistic effect on metastases inhibition, which they suggested to be induced by immune modulation by RLN, towards an immunosuppressed microenvironment. The study is original, has novel findings, very interesting and potentially clinically relevant. The manuscript is well written and the data is overall of high quality. However, prior to acceptance several issues should be addressed:

Major points:

1. The authors use models of experimental metastasis rather than spontaneous metastasis, therefore precluding investigation of changes that occur during the early stages of metastasis and the formation of a pre-metastatic niche. While they cannot be expected to change the experimental setting of their study at this stage, this limitation should be clearly stated (by using the term "experimental metastasis") and discussed. Moreover, overstatements like "We evaluated the RLN gene therapy in murine CRC and breast cancer liver metastasis models via hemi-splenic inoculation of CT26-FL3 and 4T1 cells, respectively, which mirrors the clinical scenario of liver metastasis patients.." (introduction section, line 88) should be toned down.

Answer:

We appreciate the reviewer's scientific rigor. It is absolutely correct that the liver metastasis model we used precluded investigation of changes that occur during the early stages of metastasis. As the reviewer suggested, we added more discussion (with highlight) on using the experimental liver metastasis models in the current study and its limitation in bypassing early stages of liver metastasis in **paragraph 2** under the "**Discussion**" section. The relevant content is " Notably, the experimental models used in the current study mimic the late stage of liver metastasis due to rapid tumor formation and higher metastatic rate compared with spontaneous metastasis models. And therefore, these models are well adapted for the evaluation of anti-metastasis efficacy of aHSC deactivation strategy in the established and advanced liver metastasis. However, the experimental liver metastasis models by-pass early stages of liver metastasis, including local invasion at the site of the primary tumor, gaining access to lymphatics or blood vessels and formation of a pre-metastatic niche. The more rapid tumor accumulation at the secondary site than clinical situations also possibly limits more accurate evaluation for the reference of future clinical study." We have also **added the word "experimental"** and **deleted the overstatements:** " which mirrors the clinical scenario of liver metastasis patients with the primary tumors resected" in the "**Introduction**" section, line 88-89 in the original manuscript.

2. In Figure 4, The authors use "tumor growth" (in 4d) and "metastasis inhibition" (in 4b) interchangeably, which is confusing, since in both cases they measure inhibition of the growth of experimental metastasis. This should be corrected.

Answer:

Thanks for the suggestion. To avoid the confusion, we have changed "% Metastasis Inhibition" to "Normalized tumor growth" in Fig. 4b to keep consistency with Fig. 4d. We also added the formula we used to calculate the normalized tumor growth rate in Fig. 4b to clarify. The revised figure and figure caption are also shown below. We further illustrated how we calculate the normalized tumor growth as "comparing the fold-changes over initial of the metastasis volume (V/V_0) characterized by bioluminescence intensity in the pRLN LCP treated group with those of the pGFP LCP group)" in paragraph 1 under "RLN gene therapy shows gender-biased anti-CRC liver metastasis efficacy and synergistic potential for the combination with checkpoint blockade" section of the "Results" part. All words relating to "metastasis inhibition" have been changed to "tumor growth rate" as highlighted in paragraph 1.

Figure 4b. Difference of normalized tumor growth rate between two genders after pRLN LCP treatment (n=7 for female; n=8 for male). V_R , tumor volume after pRLN LCP treatment; V_{R0} , initial tumor volume of the pRLN LCP group; V_G , tumor volume after pGFP LCP treatment; V_{G0} , initial tumor volume of the pGFP LCP group.

3. The authors speculate that the reason for the gender bias in liver metastasis is the higher physiological levels of RLN in females. In that case, treatment with exogenous RLN is not necessarily expected to be less effective in males. Nevertheless, their findings indicate a significant gender bias in inhibition of experimental metastasis. This should be discussed.

Answer:

Yes, it is correct to have a detailed discussion why the growth of experimental metastasis still differs between two genders even after the introduction of exogenous RLN. **First**, the better response towards the RLN gene therapy in female than in male mice is probably associated with **the wide gap of the endogenous RLN levels** in the liver between two genders (~5-fold difference as shown in Fig. 1f). The exogenous introduction of RLN into male mice needs to fill in the gap before achieving the similar anti-metastasis therapeutic efficacy with that in female mice. Fig. 4b also supported that the difference of the tumor growth rate between two genders was minimized with the gradual accumulation of exogenous RLN. **Second**, it was reported that progesterone and estrogen up-regulate while testosterone down-regulates RXFP1 and RXFP2 expression in the patellar tendon and lateral collateral ligament of rat's knee (*International journal of medical sciences*, 2014, 11(2): 180). Similar sex-steroid regulation of RXFP1 may also exist in the liver, which can lead to worse response towards the RLN gene therapy as well. **We have added and highlighted these discussions** at the middle of paragraph 2 under the "Discussion" section as "This gender-bias in anti-metastasis efficacy is probably also associated with the wide gap between endogenous hepatic RLN levels (~5-fold difference) in male and female mice. The difference in

therapeutic efficacy can be minimized by gradual accumulation of exogenous RLN via pRLN LCP treatments. On the other side, RLN receptor isoforms (including RXFP1) are regulated by sex-steroid as well. Progesterone and estrogen up-regulate while testosterone and androgen down-regulate RXFP1 expression, which may lead to the worse response towards the RLN gene therapy in male mice."

4. In Figure 5, the authors use FACS analysis to demonstrate that RLN treatment boosted T cell infiltration into metastatic lesions. Immunostaining of metastatic lesions to show actual infiltration would be better than only FACS analysis, especially since their point is that liver fibrosis forms a physical barrier that prevents infiltration, and RLN treatment alleviates it and enables physical entry of immune cells to a presumably less fibrotic tumor. Showing infiltration by staining is particularly important because according to the methods section, mice were not heart perfused before the metastases were harvested for analysis.

Answer:

It is absolutely correct to do the immunostaining of the metastatic lesions to more clearly show actual T cell infiltration in support of FACS analysis. As the reviewer suggested, we have done CD3⁺ T cell staining of livers with CT-26 FL3 metastasis after PBS, pRLN LCP and pRLN+pPD-L1 trap LCP treatments using anti-CD3, α -SMA, and DAPI as shown below. Microscopic analysis demonstrated

Supplementary Fig. 8 CD3⁺ T cell penetration into CRC liver metastasis. (a) Immunofluorescence staining of CT26-FL3 metastatic livers from PBS, pRLN LCP, and pRLN+pPD-L1 trap LCP treatment groups on day 13 using anti-CD3 (red), α -SMA (green), and DAPI (blue). (b) Positive ratios were quantified in 5 randomly selected fields per mouse (n=3). Bar represents 100 μ m. Significant differences were assessed using *t test*. Results are presented as mean (SD). *p < 0.05, **p < 0.01, ***p < 0.001.

aHSCs (α -SMA⁺ cells) function as a robust barrier to block the CD3⁺ T cell infiltration into metastatic lesions. The pRLN LCP treatment resulted in the reduction of aHSCs and a significant alleviation of lymphocyte exclusion (~4.3-fold increase of CD3⁺ T cell infiltration into the metastasis lesion compared with that of PBS control). The pRLN+pPD-L1 trap LCPs further enhanced T cell accumulation into metastatic lesions (more than 10-fold increase of CD3⁺ T cell infiltration into the metastasis lesion compared with that of PBS control). The figure is **highlighted** as **Supplementary Figure 8**. And the detailed illustration is **highlighted in paragraph 1** under "**pRLN therapy modifies immune microenvironment within the liver metastasis lesion**" section of the "**Results**" part.

5. Since fibrosis is a major theme of this manuscript, directly demonstrating fibrosis/resolution by staining of collagen content in the liver metastatic lesions (Masson Trichrome, Sirius Red etc.) would more directly substantiate the claim about changes in fibrosis.

Answer:

Yes, we totally agree with the reviewer. As the reviewer suggested, we have done Masson Trichrome staining for the characterization of collagen content in the liver metastatic lesions after PBS and pRLN LCP treatments as shown below. The collagen coverage in each group was further characterized by ImageJ. Clearly, it is down-regulated from 25.6% (PBS treatment) to 2.8% (pRLN LCP treatment). The

Figure 3a. α -SMA and collagen downregulation and shrunk metastatic foci after RLN gene therapy in male mice bearing CT26-FL3 liver metastasis. Immunofluorescence staining of CT26-FL3 metastatic liver from PBS and pRLN treatment groups using anti- α -SMA (green) antibody and DAPI (blue). CT26-FL3 cells were colored red. Numbers in corresponding colors indicate the average % of tumor cells and α -SMA positive cells quantified in 5 randomly selected fields per mouse (n=3). H&E and Masson's trichrome staining were performed in adjacent sections. White dotted line and arrow indicate the metastatic foci. Bars in the left two, right four panels represent 500, 200 μ m, respectively. The average % collagen (blue staining) coverage was calculated in 5 randomly selected fields per mouse (n=3).

figure is **highlighted** as Fig. 3a. And the detailed illustration is **highlighted in paragraph 1** under "RLN gene therapy reverses aHSCs to quiescence and facilitates ECM degradation " section of the "Results" part.

Minor points:

Line 44: There is a Typo ("caner" missing c); Fig. 2C legend: DiI instead of DiD; Line 246: Did the authors mean Fig. 4c? Fig. 5b: The Y axis title of the graph that shows M1/M2 ratio should be corrected; Line 300: Should be Fig. 6b and not 6a.

Answer:

We sincerely apologize for all the mistakes and much appreciate the reviewer's careful checking. We have corrected the typo mistake in line 44, updated the legend in Fig. 2C, changed the Y axis title of the graph that shows M1/M2 ratio, and corrected all other mistakes.

Referee#2: Nanoparticles, gene delivery

English has some imperfection and needs to be improved. The number of acronyms is grossly excessive affecting clarity.

Answer:

Thanks for suggestions. We have asked a native speaker to modify and improve the language for the manuscript. As to the acronyms, we have deleted all the acronyms that appeared for less than three times.

The innovation statement in line 26 and 27 “Here, we discovered an anti-fibrotic peptide, relaxin (RLN...” is contradicted by earlier publications, including Clin Med Res. 2005 Nov; 3(4): 241–249, Relaxin: Antifibrotic Properties and Effects in Models of Disease, Chishan S. Samuel (see more relevant papers at <http://onlinelibrary.wiley.com/doi/10.1111/bph.v174.10/issuetoc>) and also earlier discovery of relaxin is subsequently acknowledged in the manuscript. This statement must be modified.

Answer:

Yes, it is correct that the discovery of relaxin is not among the novelties of our study. And our statement of "we discovered an anti-fibrotic peptide, relaxin..." in line 26 and 27 can lead to misunderstanding. We are sorry about that. Indeed, **what we discovered is that intra-hepatic expression of relaxin and its receptor is under dynamic self-regulation in response to liver damages, which provides a self-repair mechanism for liver fibrosis.** To avoid the misunderstanding, **we have modified our statement as " Here, we discovered intra-hepatic scale-up of relaxin (RLN, an anti-fibrotic peptide) in response to fibrosis along with the upregulation of its primary receptor (RXFP1) on aHSCs."** at the beginning of "Abstract" section with highlight.

The nanoparticles need to be characterised far more thoroughly and the methods of production were incomplete. It is not clear how much of the gene was actually loaded (we only know how much went into the synthesis). The data in Supplementary Figure 2a were not analysed and the caption or DLS is grossly insufficient and the font size was insufficient.

Answer:

Thanks for the suggestions. We have spelled out the lipids (DOPA as 1,2-dioleoyl-sn-glycero-3-phosphate and DOTAP as 1,2-dioleoyl-3-trimethylammonium-propane) when they first appeared in paragraph 1 and 2 under "**Preparation and characterization of LCP loaded with pDNA**" section of the "**Methods**" part. We are sorry for the missing information about gene entrapment efficiency. We measured the encapsulation efficiency via fluorescence spectrometry. **The detailed method has been added and highlighted** as paragraph 3 under "**Preparation and characterization of LCP loaded with pDNA**" section of the "**Methods**" part as following: "To characterize pDNA entrapment efficiency of in LCPs, pRLN was labeled with Cy5 (Mirus LabelIT kit, Mirus Bio, Madison, WI) according to manufacturer instructions. Such Cy5-pDNA was formulated into LCPs. The final nanoparticles were dissolved in the same amount of lysis buffer (2 mM EDTA and 0.05 % Triton X-100 in pH 7.8 Tris buffer) at 65 °C for 10 min to release the entrapped pDNA⁶². The standard Cy5-pDNA solutions were prepared by diluting the original Cy5-pDNA in the same lysis buffer. Then, 100 µl of the standard and LCP solution was taken to the 96-well plate and the fluorescence intensity was measured on SpectraMax M5 plate reader (Molecular Devices) with a 620 nm excitation filter and a 670 nm emission filter." Accordingly, **the gene encapsulation efficiency was determined to be 52.2 ± 5.2% (n=3). This information has been added and highlighted in paragraph 1 under "Design of an *in situ* depot for local secretion of RLN within the liver metastatic lesion" section of the "Results" part.**

Supplementary Figure 2a confirmed the size of LCPs and indicated the spherical shape and homogenous distribution for both CaP core and the final particles. We have **highlighted this analysis in paragraph 1** under "**Design of an *in situ* depot for local secretion of RLN within the liver metastatic lesion**" section of the "**Results**" part. We have also **increased the quality of DLS figures (Supplementary Figure 2b,c) and the font size** as the reviewer suggested.

The nanoparticle production methods are insufficient, in particular they do not articulate how the targeting of nanoparticles was achieved. There is no evidence that targeting is necessary. There are no cell experiments at all, for example confirming successful targeting of cancer (there is only evidence of different liver accumulation which is unconvincing as there is no tumour outline in this figure.. There is no evidence supporting that the nanoparticle design (Fig 2 b) reflects what has been produced.

Answer:

Thanks for the suggestions. First, we have added **more details on the synthesis of DSPE-PEG-AEAA in paragraph 1** under the "**Materials**" section of "**Methods**" part as "Briefly, 4-methoxybenzoyl chloride and 2-bromoethylamine hydrobromide were mixed at room temperature for 6 h. Then, DSPE-PEG-NH₂ was added into the above solvent and stirred in an oil bath at 65-70 °C for 24 h. The final product was precipitated by ether, washed and lyophilized for further use." We did the biodistribution study on mice bearing **CT26-FL3** liver metastasis as **highlighted in the figure caption of Fig.2c**. According to Fig.2c, AEAA-conjugated LCPs had 2.4-fold higher accumulation in the metastatic liver compared with nontargeted LCPs. Whereas no difference of liver accumulation was observed between AEAA-conjugated and nonconjugated LCPs in the mice without CT26-FL3 liver metastasis according to Supplementary Fig. 3. We admit this *in vivo* distribution study is not convincing enough to illustrate the necessity of using AEAA for tumor-specific targeting. So we further **performed *in vitro* uptake study on CT26-FL3 cells**. The detailed method has been added and highlighted as "***In vitro* uptake study**" section of "**Methods**" part. As shown below, the 3.6-fold increase of intracellular uptake of Cy5-pDNA into CT26-FL3 cells by AEAA-conjugated LCPs compared with nontargeted LCPs further confirmed successful targeting of the tumor cells via AEAA. The figure has been added and highlighted as **Supplementary Fig. 4**. And the detailed analysis has been added at the end of **paragraph 1** under "**Design of an *in situ* depot for local secretion of RLN within the liver metastatic lesion**" section of the "**Results**" part.

Supplementary Fig. 4 Quantitative uptake of LCPs encapsulating Cy5-labeled pDNA into CT26-FL3 cells with or without AEAA as the targeting ligand by FACS analysis (Thermo Fisher Attune NxT).

The nanoparticle toxicity was not fully evaluated. There was no consideration as to the potential immune interactions of the treatment.

Answer:

Yes, it is correct to do more detailed analysis on the nanoparticle toxicity especially due to the usage of checkpoint blockade therapy. First, we further analyzed how LCP treatments affect white blood cell (WBC) counts and fractions 2h, 24h, and 7 days after the final injections. In addition to the previous result, we further found a sudden increase of WBC counts along with significant fluctuations of lymphocyte and neutrophil fractions in WBCs 2h after the free RLN injection compared with the PBS control. **In contrast, pRLN LCP treatment did not induce WBC changes either 2h or 24h after administration** as shown below (**Supplementary Fig. 10a**), which suggested **lower immunogenicity of the nanoparticles than the recombinant peptide**. This inflammatory effect of recombinant RLN is mild and transient as evidenced by the disappearance of significant changes in WBC counts and cell fractions 24h after the treatment as shown below (**Supplementary Fig. 10b**). We also analyzed cell fractions of WBCs from each treatment group 7 days after the last treatment. The significant up-regulation of lymphocyte and down-regulation of neutrophil fractions in WBCs by pRLN and pRLN+pPD-L1 trap as shown below (**Supplementary Fig. 10c**) supported less expansion of the liver metastasis after these treatments.

Immune checkpoint inhibitors are known to associate with immune-related adverse events (irAEs) driven by unmasking self-reactive immune cells. IL-17 and Th17 cells are highly up-regulated in inflammatory tissues of autoimmune diseases. Hence, the ratio of Th17 cells can be used as a parameter to monitor the irAEs of checkpoint inhibitor immunotherapy. A previous study has confirmed elevated Th17 cells in the spleen after systemic anti-PD-L1 mAb treatment (*Nature communications*, 2018, 9(1): 2237). In contrast, **neither pPD-L1 trap nor pRLN+pPD-L1 trap treatment groups showed any significant increases of Th17 cells in the spleen compared with the PBS control** as shown below (**Supplementary Fig. 11**). Lipopolysaccharide (LPS) is known to promote the generation of Th17 cells (*Immunology letters*, 2015, 165(1): 10-19). Therefore, 24h after a single dose of LPS treatment (50µg, i.p.), the spleen was resected and used as a positive control. As expected, flow cytometry analysis confirmed LPS injection increased Th17 cells by 2-fold while the **pRLN+pPD-L1 trap treatment did not stimulate the up-regulation of Th17 cells compared with the PBS control** as shown below (**Supplementary Fig. 12**). The low adverse effects are presumably associated with local and transient expression of PD-L1 trap protein within the metastatic lesions. We have added and highlighted the additional analysis under the "**Toxicity evaluation for different treatments and blood chemistry analysis**" section of the "**Results**" section. The newly produced **Supplementary Fig. 11** and **Supplementary Fig. 12** have been added and highlighted in the supporting information.

Supplementary Fig.10 Complete blood count and blood chemistry analysis of CT26-FL3 liver metastasis bearing mice after various treatments 2h (a), 24 h (b) or 7 days (c) after the last injection (n = 4). Blue dotted lines represent the normal range of each indicator. Significant differences were assessed using *t* test. Results are presented as mean (SD). **p* < 0.05, ***p* < 0.01, ****p* < 0.001, n.s., no significance. The *p* values of individual groups in (c) were calculated by comparing to the PBS control.

Supplementary Fig. 11 Th17 cells in the spleen of CT26-FL3 liver metastasis bearing mice. **(a)** Immunofluorescence staining of Th17 cells (CD4⁺IL-17⁺ T cells) in the spleen from PBS, pPD-L1 trap, pRLN, and pRLN+pPD-L1 trap treatment groups on day 13 using anti-CD4 (red), anti-IL-17 (green), and DAPI (blue). **(b)** %Th17 cells in CD4⁺ cells were quantified in 5 randomly selected fields per mouse (n=3). Bar represents 50 μm. Significant differences were assessed using t test. Results are presented as mean (SD). n.s., no significance.

Supplementary Fig. 12 Flow cytometry analysis of Th17 cells in the spleen of C57BL/6 male mice bearing KPC liver metastasis receiving various treatments on day 14. **(a)** Representative scatter plots and **(b)** Quantitative analysis of CD4⁺IL-17⁺ cells in the spleen (n=4). Significant differences were assessed using t test. Results are presented as mean (SD). ***p < 0.001, n.s., no significance.

Reviewer #3 (Remarks to the Author):

In this manuscript the authors described the role of relaxin (RLN) peptide in liver metastasis. Enforced expression of RLN in liver deactivated the aHSCs and changes in tumor immune microenvironment resulted in controlling the liver metastasis. The authors utilized anisamide conjugated targeted LCP nanoparticles to encapsulate RLN encoded plasmid DNA to target tumor cells and aHSCs. Additionally, nanoparticles containing both RLN and PD-L1 pDNAs prolonged the survival of mice in breast and colorectal cancer liver metastasis models. The subject is interesting and the results are significant. However, the manuscript lacks novelty and the following issues need to be considered prior to publication in nature communications.

Major comments:

1. RXFP is widely expressed in all peripheral tissues of male and female species with different roles in different organs and not restricted to the liver. Additionally, several other RXFP family receptors are expressed in the human body. Interestingly, RLN has high affinity ligand for all RXFP receptors. The authors did not consider these issues and the expression of RLN in other organs might affect their functions? These issues need to be considered and addressed in the manuscript.

Answer:

We appreciate the reviewer's scientific rigor. It is correct that RXFP1 as well as other subtypes of relaxin receptors are widely expressed in all peripheral tissues with different roles and not restricted to the liver. And these receptors all have high binding affinity with RLN. **We have mentioned the wide distribution of RXFP1 throughout the body**, which will compromise the therapeutic efficacy of systemically administrated RLN peptide **at the end of paragraph 3** under the "**Introduction**" section. To be clearer, **we re-wrote and highlighted this part as** "systemic administrated RLN usually displays compromised efficacy due to the short half-life (~10 min). The widely distributed RXFP1, particularly in the reproductive system, also reduces the RLN amount arriving the targeted site and induces possible side-effects.". We also added and highlighted similar statements in **paragraph 2** under the "**Design of an *in situ* depot for local secretion of RLN within the liver metastatic lesion**" section of the "**Results**" part.

The advantage of our delivery system is that it allows local expression of RLN within the metastatic lesions. As shown in **Fig. 2e**, highly up-regulated RLN level **was only observed in the metastatic liver** after i.v. injection of pRLN LCP and **no significant difference of RLN expression was found in other major organs including heart, lung, spleen, and kidney**. The slightly increased RLN level in the serum might be associated with leaking of excessive RLN expressed in the metastatic liver. Nevertheless, **the level (0.2 ± 0.1 pg/mL) is far below the toxicity limit (~13 ng/mL)** (*BMC pregnancy and childbirth*, 2016, 16(1): 260). More specifically, we further found the delivered gene (we used pGFP as the marker gene) was selectively expressed within the metastasis foci but not in the metastasis-free area in the liver. The figure has been added and highlighted as **Supplementary Figure 6**. These results demonstrate **the delivery system we used allows for *in situ* expression in the liver metastatic lesion with minimal off-target effects**.

Fig. 2e RLN peptide expression in the major organs and serum 2 days after the final pDNA LCP injection (n=5). Significant differences were assessed using t test. Results are presented as mean (SD). *p < 0.05, **p < 0.01, ***p < 0.001, n.s., not significant.

Supplementary Fig. 6 GFP expression in the CT26-FL3 metastasis-free area of the liver after 3 injections of pGFP LCP into CT26-FL3 liver metastasis bearing mice. GFP and tumor cells are colored green and red, respectively. Immunofluorescence staining using anti- α -SMA (cyan) and DAPI (blue) were further performed. Bar represents 100 μ m.

2. It is very interesting that anisamide can target HSCs? Are they expressing the sigma receptors? The total expression of protein in the liver could be from tumor cells only? If so how the RLN expression in tumors can control liver metastasis?

Answer:

Fig. 2a Expression of sigma-1 receptor on aHSCs. Immunofluorescence staining of CT26-FL3 metastatic liver from PBS group using anti- α -SMA (green), anti-sigma-1 receptor (red) antibodies, and DAPI (blue). CT26-FL3 cells were colored Cyan. Bars in the middle and right panels represent 100 and 50 μ m, respectively.

Yes, aminoethyl anisamide can target the activated HSCs. As shown in Fig. 2a (also shown above), we

Supplementary Fig. 5 GFP expression in the liver with or without CT26-FL3 metastatic foci. **(a)** GFP expression on day 1, 2, 4, 6, and 8 after 3 injections of pGFP LCP into CT26-FL3 liver metastasis bearing mice. GFP and tumor cells are colored green and red, respectively. Immunofluorescence staining using anti- α -SMA (cyan) and DAPI (blue) were further performed. Bar represents 100 μ m. **(b)** Quantification of % tumor cells, % aHSCs, and % other cells in GFP⁺ cells (upper panel), as well as %GFP⁺ cells in tumor cells and aHSCs (characterized by α -SMA⁺) (lower panel) in 5 randomly selected fields per mouse (n=3). Significant differences were assessed using t test. Results are presented as mean \pm SD. **p < 0.01, ***p < 0.001.

found significantly increased expression of the sigma-1 receptor in the metastatic liver compared to the healthy control. More specifically, Sig-1R primarily located surrounding the RFP-positive CT26-FL3 cells or α -SMA-positive aHSCs instead of overlapping with the cytosolic RFP or α -SMA. **The statement is highlighted in paragraph 1** under the "**Design of an *in situ* depot for local secretion of RLN within the liver metastatic lesion**" section of the "**Results**" part. We monitored the intra-hepatic gene expression profile via the marker gene, green fluorescence protein plasmid (pGFP). As shown in Supplementary Fig. 5a (also shown above), GFP was mainly expressed by tumor cells and aHSCs. And

tumor cells showed ~2-fold greater gene expression capacity than aHSCs. Different from GFP (an intracellular protein), **RLN is a secretory peptide**. The gene encodes for RLN contains an endogenous signal motif that commits the peptide for extracellular secretion (*Trends in Endocrinology & Metabolism*, 2002, 13(8): 343-348). The close location of aHSC surrounding tumor cells within metastatic lesions facilitates the binding of secreted RLN with RXFP1 on aHSCs via either paracrine or autocrine pattern. We have added and highlighted the illustration in **paragraph 2** under the "**Design of an *in situ* depot for local secretion of RLN within the liver metastatic lesion**" section of the "**Results**" part.

3. Can this therapy be applicable to other cancer metastasis models?

Answer:

Yes, we further tested the effect of RLN gene therapy and its combination with PD-L1 blockade in pancreatic cancer liver metastasis model because the liver is also the most common metastatic site for pancreatic cancer. And recent studies demonstrated highly fibrotic stroma in liver metastases of pancreatic ductal adenocarcinoma (PDAC, the most common type of pancreatic cancer) and its critical role to sustain metastatic tumor growth (*Nature cell biology*, 2016, 18(5): 549; *Clinical Cancer Research*, 2015, 21(15): 3561-3568.). KPC, a genetically engineered mouse model with mutations in proto-oncogene K-Ras and tumor-suppressor p53 mutations, is a clinically relevant model of PDAC. Therefore, experimental liver metastasis of a primary tumor cell line generated from KPC mice was used to test the effect of RLN gene therapy and its combination with PD-L1 blockade immunotherapy. As observed in CT26-FL3 liver metastasis, **KPC liver metastasis was refractory to PD-L1 blockade therapy, while the combination of pRLN and pPD-L1 trap efficiently reduced the metastatic burden (Fig. 6a, 6b, and 6c)**. pRLN LCP treatment alone could substantially slow down the progression of metastatic tumor growth during the dosing period. However, the metastasis developed quickly once the treatment stopped, leading to only slightly prolonged survival (**Fig. 6d**). In contrast, pRLN+pPD-L1 trap LCP treatment induced tumor regression in 50% of the KPC metastasis-bearing mice (**Fig. 6b**) and significantly extended the median survival in comparison with PBS control and pRLN treatment groups (**Fig. 6d**). We have added and highlighted the relevant data in **Fig 6a-d** (also shown below). The detailed analysis is added and highlighted as **paragraph 1** under "**Test in other liver metastasis models**" section of the "**Results**" part.

Fig. 6 RLN and PD-L1 blockade gene-immune therapy on KPC and 4T1 liver metastases. **(a)** KPC tumor inoculation and treatment scheme. **(b)** Representative in vivo bioluminescence imaging of mice bearing KPC liver metastasis receiving various treatments on day 3, 7, 11 post tumor inoculation. **(c)** Quantification analysis of KPC liver metastasis burden by bioluminescence intensity ($n=6$). **(d)** Survival curves of mice bearing KPC liver metastasis in each treatment group ($n=6$). Significant differences were assessed in c using t test, in d using log rank test. Results are presented as mean (SD). * $p < 0.05$, ** $p < 0.01$, *** $p < 0.001$, n.s., not significant.

4. If the RLN has significant role in liver metastasis as mentioned in the introduction, why there is no difference in mice's survival between free RLN and pRLN treated groups? Here only combination of pPD-L1 and pRLN showed significant difference in metastasis inhibition. The authors should justify the role of RLN alone in aHSCs deactivation followed by metastasis inhibition.

Answer:

Indeed, there was no difference in mice's survival among PBS, pGFP, pPD-L1, and free RLN peptide treatment groups. **But pRLN LCP treatment could significantly increase the median survival** by 25% compared with these groups. The combination of pPD-L1 and pRLN treatment further increased the median survival by 2-fold compared with the PBS group and increased the survival by 50% compared with pRLN LCP treatment alone. We have revised and highlighted **Fig. 4e** to show the significant differences more clearly.

Fig. 4e Mice survival curves in each treatment group (n=8). Significant differences were assessed using log rank test. Results are presented as mean (SD). ***p < 0.001, n.s., not significant.

5. The authors should justify the use of pDNA due to their adverse immune reactions, which is an important aspect in clinical translation.

Answer:

Yes, it is correct to do more detailed toxicity analysis especially due to the usage of checkpoint blockade therapy. First, we further analyzed how LCP treatments affect white blood cell (WBC) counts and fractions 2h, 24h, and 7 days after the final injections. In addition to the previous result, we further found a sudden increase of WBC counts along with significant fluctuations of lymphocyte and neutrophil fractions in WBCs 2h after the free RLN injection compared with the PBS control. **In contrast, pRLN LCP treatment did not induce WBC changes either 2h or 24h after administration** as shown below (**Supplementary Fig. 10a**), which suggested **lower immunogenicity of the nanoparticles than the recombinant peptide**. This inflammatory effect of recombinant RLN is mild and transient as evidenced by the disappearance of significant changes in WBC counts and cell fractions 24h after the treatment as shown below (**Supplementary Fig. 10b**). We also analyzed cell fractions of WBCs from each treatment group 7 days after the last treatment. The significant up-regulation of lymphocyte and down-regulation of neutrophil fractions in WBCs by pRLN and pRLN+pPD-L1 trap as shown below (**Supplementary Fig. 10c**) supported less expansion of the liver metastasis after these treatments.

Immune checkpoint inhibitors are known to associate with immune-related adverse events (irAEs) driven by unmasking self-reactive immune cells. IL-17 and Th17 cells are highly up-regulated in inflammatory tissues of autoimmune diseases. Hence, the ratio of Th17 cells can be used as a parameter to monitor the irAEs of checkpoint inhibitor immunotherapy. A previous study has confirmed elevated Th17 cells in the spleen after systemic anti-PD-L1 mAb treatment (*Nature communications*, 2018, 9(1): 2237). In contrast, **neither pPD-L1 trap nor pRLN+pPD-L1 trap treatment groups showed any significant increases of Th17 cells in the spleen compared with the PBS control** as shown below (**Supplementary Fig. 11**). Lipopolysaccharide (LPS) is known to promote the generation of Th17 cells (*Immunology letters*, 2015, 165(1): 10-19). Therefore, 24h after a single dose of LPS treatment (50µg, i.p.), the spleen was resected and used as a positive control. As expected, flow cytometry analysis confirmed LPS injection increased Th17 cells by 2-fold while the **pRLN+pPD-L1 trap treatment did not stimulate the up-regulation of Th17 cells compared with the PBS control** as shown below (**Supplementary Fig. 12**). The low adverse

effects are presumably associated with local and transient expression of PD-L1 trap protein within the metastatic lesions. We have added and highlighted the additional analysis under the "**Toxicity evaluation for different treatments and blood chemistry analysis**" section of the "**Results**" part. The newly produced **Supplementary Fig. 11** and **Supplementary Fig. 12** have been added and highlighted in the supporting information.

Supplementary Fig.10 Complete blood count and blood chemistry analysis of CT26-FL3 liver metastasis bearing mice after various treatments 2h (a), 24 h (b) or 7 days (c) after the last injection (n = 4). Blue dotted lines represent the normal range of each indicator. Significant differences were assessed using *t test*. Results are presented as mean (SD). *p < 0.05, **p < 0.01, ***p < 0.001, n.s., no significance. The p values of individual groups in (c) were calculated by comparing to the PBS control.

Supplementary Fig. 11 Th17 cells in the spleen of CT26-FL3 liver metastasis bearing mice. **(a)** Immunofluorescence staining of Th17 cells (CD4⁺IL-17⁺ T cells) in the spleen from PBS, pPD-L1 trap, pRLN, and pRLN+pPD-L1 trap treatment groups on day 13 using anti-CD4 (red), anti-IL-17 (green), and DAPI (blue). **(b)** %Th17 cells in CD4⁺ cells were quantified in 5 randomly selected fields per mouse (n=3). Bar represents 50 μ m. Significant differences were assessed using t test. Results are presented as mean (SD). n.s., no significance.

Supplementary Fig. 12 Flow cytometry analysis of Th17 cells in the spleen of C57BL/6 male mice bearing KPC liver metastasis receiving various treatments on day 14. **(a)** Representative scatter plots and **(b)** Quantitative analysis of CD4⁺IL-17⁺ cells in the spleen (n=4). Significant differences were assessed using t test. Results are presented as mean (SD). ***p < 0.001, n.s., no significance.

6. pRLN therapy alone may not be efficient in some tumors (4T1) compared to CT26? Why there was significant difference in therapeutic outcome between the two tumor models with pRLN, where RLN is critical factor in stromal microenvironment? What is the effect of combination therapy on 4T1 metastasis model?

Answer:

Indeed, pRLN LCP treatment alone seems to **show better anti-metastasis efficacy on 4T1 than CT26 liver metastasis**. The anti-metastasis efficacy was maintained for another week after pRLN LCP treatments (**Fig. 6h**), which led to ~2-fold increase of survival compared with PBS control (**Fig. 6i**). In contrast, pRLN LCP treatment alone only slightly prolonged the median survival by 25% compared with PBS control in CT26 liver metastasis. The better therapeutic efficacy of pRLN LCP on 4T1 liver metastasis (**female mice**) than on CT26 liver metastasis (**male mice**) is **probably due to the better response to RLN therapy in female mice**. As shown in Supplementary Fig. 8, pRLN LCP treatment alone on female mice bearing CT26 liver metastasis also showed similar anti-metastasis efficacy as on female mice bearing 4T1 liver metastasis. Because breast cancer and its liver metastasis are rare in male, we did not use male mice to study the anti-metastasis efficacy on 4T1 liver metastasis. And since we have proven the effect of combinational therapy on both CT26 and KPC liver metastasis models, we feel that it is not necessary to repeat the study on the 4T1 liver metastasis model, especially when pRLN LCP treatment alone has already displayed good anti-metastasis efficacy.

7. Reports also suggesting that RLN expression can increase the risk of prostate cancer and promotes angiogenesis. The authors need to carefully consider and discuss this.

Answer:

Thanks for the suggestions. We have added and highlighted "It is also noteworthy that despite the positive effect of the RLN gene therapy reported here, previous studies on the prostate cancer demonstrated that tumor cells overexpressing H2-RLN had a greater xenograft tumor volume than wild-type controls due to increased angiogenesis. Mutated H2-RLN, which functioned as an RXFP1 antagonist, impaired prostate tumor growth. Nevertheless, the usage of immune-deficient mouse models in these studies might overlook RLN-related immune involvement, which has been proven crucial for the RLN gene therapy here. Furthermore, in contrast with high RXFP1 expression on the prostate cancer cells in these studies, the lack of RXFP1 expression in both CRC and breast cancer cells (**Supplementary Fig.14**) resulted in unresponsiveness of tumor cells to RLN" at the end of **paragraph 2** of "**Discussion**" section. We also added **Supplementary Fig.14** in the supporting file (also shown below).

Supplementary Fig. 14 RXFP1 expression in human CRC and breast cancers. **(A), (C)**. mRNA level of RXFP1 in CRC or breast cancer patient samples cited from TCGA database. **(B), (D)**. Representative tumor tissue of CRC cancer patient sample (CRC-2948) or breast cancer patient sample (breast cancer-2174) cited from Swedish-based program-The Human Protein Atlas.

REVIEWERS' COMMENTS:

Reviewer #1 (Remarks to the Author):

In the revised manuscript the authors have fully addressed all my comments.

The manuscript is much improved and it is interesting and novel,

I recommend publication of this study.

Reviewer #2 (Remarks to the Author):

This reviewer is now satisfied with the changes.

Reviewer #3 (Remarks to the Author):

The authors have made dramatic revisions and addressed my concerns. The paper is now ready for publication.

- Dan Peer